# QPrompt-R1: Real-Time Reasoning for Domain Generalized Semantic Segmentation via Group-Relative Query Alignment

**Fengyuan Lu**[1], **Zixuan Duan**[1], **Xunzhi Xiang**[1], **Zhicheng Zhang**[2], **Wenbin Li**[1], **Yang Gao**[1], **Qi Fan**[1,†]

[1]Nanjing University    [2]Shenzhen International Graduate School, Tsinghua University

fanqi@nju.edu.cn

## Abstract

Deploying semantic segmentation in driving and robotics requires both real-time inference and robustness to domain shifts, formalized as *Real-Time Domain-Generalized Semantic Segmentation* (RT-DGSS), a challenge not fully addressed. Existing methods treat real-time (RT) inference and domain generalization (DG) separately, with DG improving robustness but lacking real-time performance. To tackle the RT-DGSS problem, we identify that the bottleneck in DG is the prediction head, not the backbone. We introduce **QPrompt-R1**, a real-time *Query-Prompt* architecture based on the powerful VFM backbone. QPrompt-R1 integrates reasoning by injecting learnable queries into the final transformer block, leveraging contextual learning to enhance segmentation performance under domain shifts while maintaining real-time inference. To further optimize reasoning without extra inference cost, we introduce a *Group Relative Query Alignment (GRQA)* training objective, which strengthens the relationship between queries and image tokens through group-relative advantage supervision, unlocking the domain generalization potential of VFMs. QPrompt-R1 achieves **54 FPS**, delivering strong performance in synthetic-to-real transfer, real-to-real generalization, and robustness under adverse conditions. GRQA functions as a *plug-and-play* module, improving DGSS methods such as REIN (+1.2) and SoMA (+0.6) without introducing inference-time overhead. The code is available at QPrompt-R1.

## 1 Introduction

Semantic segmentation in autonomous driving and robotics requires both *real-time inference* and *robustness to distribution shifts*. Real-time segmentation supports safety-critical tasks like obstacle avoidance in autonomous vehicles (Grigorescu et al., 2020; Holder & Shafique, 2022) and robot navigation (Guan et al., 2022). Robustness ensures generalization across environments, handling weather, lighting, and terrain variations (Gao et al., 2024).

Recent research has treated *real-time performance* and *robustness under distribution shifts* as separate optimization goals. Current DGSS approaches (Wei et al., 2024; Yun et al., 2025; Bi et al., 2024; Zhang & Robby T., 2025) mainly focus on exploit VFMs to improve robustness. In contrast, RTSS methods focus on designing novel architecture to optimize the accuracy-latency trade-off for high-speed performance. Despite their advances, DGSS methods face high computational costs, hindering real-time deployment, while RTSS methods sacrifice domain adaptability due to fixed class embeddings. This trade-off underscores a critical gap in current research: *Why do existing methods fail to effectively combine RT and DGSS?*

State-of-the-art DGSS methods (Wei et al., 2024; Yun et al., 2025; Bi et al., 2024; Zhang & Robby T., 2025; Liu et al., 2025) leverage VFMs to enhance domain robustness. However, we identify that the speed bottleneck lies not in the VFMs backbone, but in the sophisticated segmentation head. Most DGSS models, based on a query-based head, rely on a complex segmentation head with a pixel encoder and transformer decoder, using the VFM only in the backbone as fig. 1a. Replacing

---

†Corresponding author.

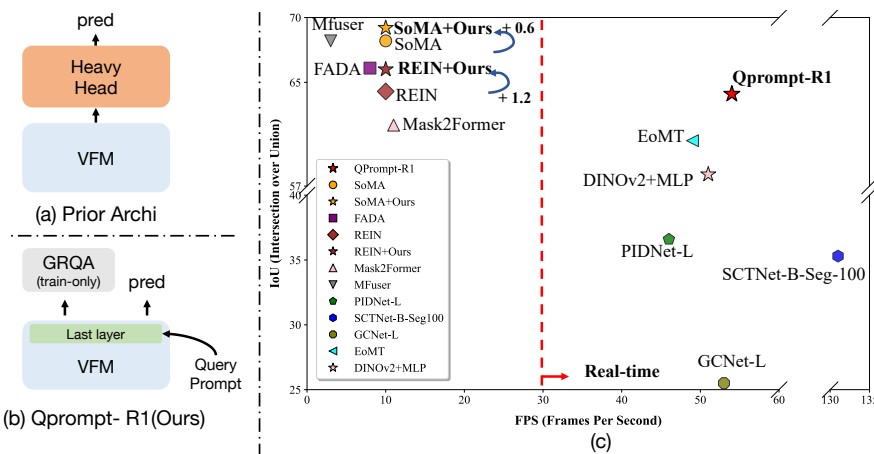

Figure 1: (a) Prior DGSS architectures rely on heavy segmentation heads. (b) QPrompt-R1 integrates query prompts at the final VFM layer with train-only GRQA, enabling efficient alignment and generalization. (c) Speed–accuracy trade-off under *GTAV→{Citys, BDD, Map}*. QPrompt-R1 achieves an optimal balance, while GRQA remains scalable and further boosts performance.

the query-based head with a simpler MLP head, commonly used in RTSS methods, results in a significant speed boost (FPS), as shown in Fig 1(c), highlighting the query-based head as the time bottleneck of DGSS. On the other hand, most RTSS methods (Xu et al., 2022; 2023; Yang et al., 2025) rely on CNN architectures, which cannot effectively leverage the generalization power of VFMs. Moreover, they lack the ability for in-context adaptive learning, as they learn a fixed set of class embeddings, while the query-based approach can fuse with image token context for adaptive learning, effectively mitigating the effects of domain shifts. Therefore, we need a novel architecture for RT-DGSS problem.

Recent advances in reinforcement learning (RL) have introduced new paradigms for post-training large models, enabling alignment with complex objectives beyond supervised fine-tuning. In language models, DeepSeekMath introduced Group Relative Policy Optimization (GRPO) (Shao et al., 2024), which replaces the critic with group-relative baselines, pushing the boundaries of mathematical reasoning. GRPO has also been applied to vision tasks (Pan et al., 2025; Huang et al., 2025; Yu et al., 2025), enhancing the capabilities of vision models.

To address the RT-DGSS problem, we introduce the **QPrompt**(Fig. 1b), a lightweight, single-layer prompting mechanism integrated at the final Transformer layer in VFM, inspired by prior query-based model and prompt/token tuning for vision models (Jia et al., 2022; Kerssies et al., 2025). Concretely, QPrompt injects a small set of learnable queries only into the final transformer block, approximating query-based decoding without a multi-layer decoder stack. QPrompt retains the adaptive nature of query-based methods, with minimal computational overhead, adding only $K$ extra tokens in the final transformer layer. However, a single layer interaction between image tokens and queries is insufficient for robust performance. Inspired by GRPO (Shao et al., 2024), we propose **Group-Relative Query Alignment (GRQA)** to enhance query-prompted domain-generalized reasoning. Our key insight is to train all queries within a class-specific group, allowing multiple queries to acquire segmentation competence and mitigate failures under domain shifts. Unlike Hungarian matching, which assigns a single query to each ground-truth mask, GRQA leverages *group-relative advantages* to enable mutual supervision and jointly optimize all queries. This method is fully supervised during training, with all auxiliary components disabled at test time, ensuring *no* inference-time overhead. As a plug-and-play module, GRQA can be easily integrated with existing DGSS methods like REIN and SoMA, providing performance improvements without increasing inference-time cost.

Building upon this, we propose **QPrompt-R1**, a real-time, domain-generalized semantic segmentation model (QPrompt) enhanced with Group-Relative Query Alignment (GRQA) optimization for improved reasoning. QPrompt-R1 achieves a sustained inference speed of **54 FPS**, demonstrating strong synthetic-to-real transfer, real-to-real generalization under adverse conditions. QPrompt-R1 performance pushes the frontier of real-time systems while narrowing the performance gap to domain-generalized semantic segmentation (DGSS) methods.

We make the following contributions:

- We highlight Real-Time Domain-Generalized Semantic Segmentation (RT-DGSS) as an important and practical research challenge, addressing both robustness to domain shifts and real-time inference efficiency.

- We propose QPrompt-R1, a real-time and robust semantic segmentation model, along with a plug-and-play GRQA training strategy designed to enhance the model's generalization. QPrompt-R1 achieves a balanced trade-off between performance and efficiency.

- Group-Relative Query Alignment (GRQA) is a generalizable approach that can be combined with existing DGSS methods to push the limit of domain generalization.

## 2 RELATED WORKS

**Domain Generalized Semantic Segmentation.** Domain-generalized semantic segmentation (DGSS) aims to maintain high accuracy under distribution shifts from diverse urban layouts, weather, and lighting conditions. Early methods used style transfer, feature normalization, and adversarial alignment (e.g., (Zhou et al., 2022c), (Chattopadhyay et al., 2023), (Kim et al., 2022), (Cho et al., 2023), (Pan et al., 2018)) to learn domain-invariant representations. Recently, vision foundation models (VFMs) have become powerful backbones for DGSS, with methods like REIN (Wei et al., 2024), FADA (Bi et al., 2024), and SoMA (Yun et al., 2025) refining VFMs through parameter-efficient tuning, frequency-domain adaptation, or low-rank adjustments. Other approaches, such as MFuser (Zhang & Robby T., 2025), combine VFMs with vision–language models (VLMs) to exploit multimodal priors. Despite these advances, most DGSS methods focus on robustness, neglecting real-time applicability, which is crucial for safety-critical tasks like autonomous driving and robotics. We contend that both robustness and efficiency must be jointly addressed. While previous methods implicitly touch upon these aspects, we establish Real-Time Domain-Generalized Semantic Segmentation (RT-DGSS) as a distinct research setting to rigorously evaluate the trade-off between inference speed and domain generalization.

**Real-Time Semantic Segmentation.** Real-time semantic segmentation is crucial for applications such as autonomous driving and robotics, where fast, reliable pixel-level prediction is required. Early work mainly relies on lightweight CNN designs to balance accuracy and efficiency, e.g., BiSeNet (Yu et al., 2018) decouples spatial detail and context with a dual-path architecture, and PIDNet (Xu et al., 2022) introduces a three-branch structure to explicitly model boundary cues. More recently, transformer/hybrid designs have also achieved strong accuracy–latency trade-offs. RTFormer (Wang et al., 2022) proposes an efficient dual-resolution transformer with GPU-friendly attention for real-time segmentation. SeaFormer (Wan et al., 2023) develops mobile-friendly axial-transformer backbones coupled with lightweight segmentation heads, targeting low-latency deployment on edge devices. Next-ViT (Li et al., 2022) introduces an efficient deployment-oriented backbone that offers a strong latency–accuracy trade-off for dense prediction. Despite these advances, prior RTSS methods primarily optimize latency and in-domain accuracy, leaving robustness under domain shifts largely unexplored. In contrast, we target Real-Time Domain Generalized Semantic Segmentation (RT-DGSS), which preserves efficiency while improving cross-domain generalization.

**RL-based Post-training and GRPO.** Reinforcement learning (RL)(Ouyang et al., 2022)(Schulman et al., 2017)(Guo et al., 2025) has become essential for post-training large models, aligning them with objectives beyond supervised fine-tuning. In language models, DeepSeekMath introduced Group Relative Policy Optimization (GRPO) (Shao et al., 2024), replacing the critic with group-relative baselines to enhance mathematical reasoning. Similar approaches were applied to vision: (Pan et al., 2025) proposed Group Relative Query Optimization (GRQO) for denser query supervision in vision transformers, (Yu et al., 2025) applied GRPO to multimodal tasks, and Vision-R1 (Huang et al., 2025) demonstrated improvements in multimodal reasoning. However, group-relative optimization for dense prediction tasks like semantic segmentation remains underexplored. We address this gap with **Group Relative Query Alignment (GRQA)**, which adapts GRPO-style rewards for query–image alignment in segmentation transformers without additional inference cost.

## 3 METHODOLOGY

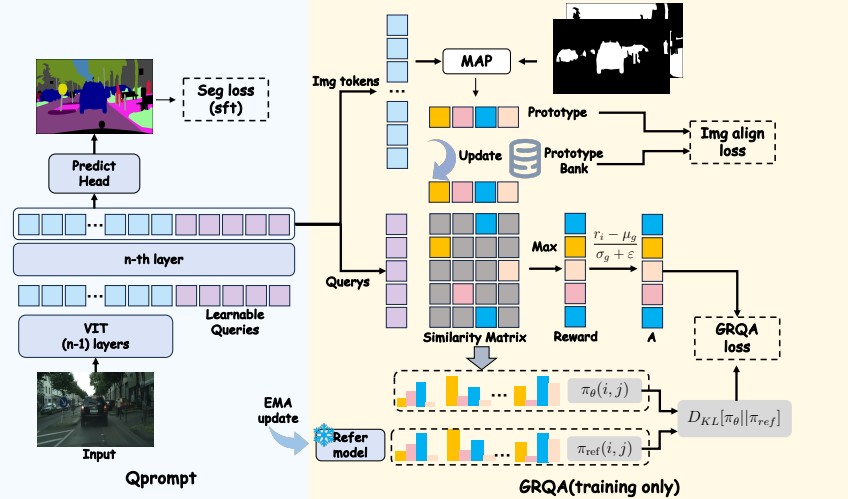

Figure 2: Overview of QPrompt-R1. Left: QPrompt employs a ViT backbone, injecting learnable queries only in the final layer to capture context, followed by a prediction head. Right: The GRQA module builds a prototype bank and computes group-relative advantages to optimize queries, enhancing domain-generalized reasoning. GRQA is training-only and incurs no inference overhead.

## 3.1 PRELIMINARIES: GROUP-RELATIVE POLICY OPTIMIZATION (GRPO)

Group-Relative Policy Optimization (GRPO) (Shao et al., 2024) is a simplified policy-optimization framework that replaces the value function with a group-relative advantage. Given a group of outputs, each output's advantage is computed by subtracting the group mean reward: $\hat{A}_i = r_i - \frac{1}{G}\sum_{j=1}^{G} r_j$ The policy is updated by maximizing a PPO-style objective augmented with KL regularization:

$$\mathcal{L}_{\text{GRPO}} = \mathbb{E}\left[\min\left(\rho_i \hat{A}_i,\ \text{clip}(\rho_i, 1-\epsilon, 1+\epsilon)\hat{A}_i\right)\right] - \beta\, D_{\text{KL}}[\pi_\theta \,\|\, \pi_{\text{ref}}], \tag{1}$$

where $\pi_\theta$ denotes the policy probability, $\rho_i = \pi_\theta/\pi_{\text{ref}}$ the importance ratio between the current and reference policies, and the KL divergence term encourages conservative updates by keeping the learned policy close to a reference model. These designs contain two central ideas—group relative advantages and conservative updates, which form the basis of GRQA objective.

## 3.2 QPROMPT FOR SEMANTIC SEGMENTATION

Since VFMs are pretrained on large-scale datasets and demonstrate strong generalization (Oquab et al., 2023; Kirillov et al., 2023), we aim to exploit their inherent strengths with a simple architecture and training-only strategy rather than heavily relying on complex segmentation heads. Motivated by query-based heads (Cheng et al., 2022), which allow queries to adaptively interact with image tokens for in-context learning, we introduce a real-time architecture, **QPrompt** (as illustrated in Fig. 2), to preserve the advantages of query-based approaches while reducing computational overhead. Formally, we define the VFM backbone as a sequence of $L \geq 2$ Transformer blocks, $\{\mathcal{B}_1, \ldots, \mathcal{B}_L\}$. Let $x_\ell \in \mathbb{R}^{N \times d}$ denote the token sequence after block $\ell$. An input image is first processed by the backbone through the initial $L-1$ blocks, producing the intermediate tokens $x_{L-1}$. At the final block, $K$ learnable queries $Q \in \mathbb{R}^{K \times d}$ are concatenated with the $(L-1)$-th output to form the augmented tokens:

$$\tilde{x}_{L-1} = [Q, x_{L-1}], \tag{2}$$

which are then fed into the last Transformer block $\mathcal{B}_L(\cdot)$. The block outputs the refined queries and updated image tokens:

$$[Q_L, x_L] = \mathcal{B}_L(\tilde{x}_{L-1}). \tag{3}$$

Following standard query-based decoding, each refined query $Q_L$ predicts class logits and generates per-pixel predictions by attending back to image tokens $x_L$. To recover fine-grained boundaries from patchified VFM features, QPrompt employs a lightweight upsampling head, consisting of two learnable transposed-convolution layers. In this way, queries serve as adaptive class embeddings that directly produce the segmentation map without requiring an additional decoder stack.

By injecting queries at the last block, QPrompt approximates the role of both pixel encoder and transformer decoder in conventional query-based methods within a single layer interaction. Previous query-based model (Cheng et al., 2022) employs a pixel encoder followed by a multi-layer decoder, with complexity $\mathcal{O}(M(N+K)^2d)$, $M$ is the number of decoder layers. QPrompt reduces to a single Transformer block over $N+K$ tokens ($\mathcal{O}((N+K)^2d)$), retaining properties of query-based methods and improve inference speed. Unlike EoMT (Kerssies et al., 2025), whose complex mask attention and annealing introduce a detrimental train-test process discrepancy, QPrompt is a simpler, consistent architecture that ensures train-test parity for stable generalization.

### 3.3 GROUP RELATIVE QUERY ALIGNMENT

While QPrompt leverages query-based methods to reduce computational overhead, a single-layer interaction between image tokens and queries may be insufficient for robust generalization, particularly for handling domain shifts. To address this, we propose the **Group Relative Query Alignment (GRQA)** strategy, which enhances query interactions without increasing inference cost. During training, Hungarian matching (Cheng et al., 2021) assigns only one query per class, relegating the others to the background. This results in only one query per class being supervised, preventing the training of multiple alternative queries to handle domain shift issues (Wen & Li, 2024). To enable efficient and stable query optimization, we adopt a momentum-updated prototype bank as standard class anchors widely used in prior segmentation works (Tang et al., 2025; Zhou et al., 2022b).

**Prototype Bank.** To provide stable, class-specific references for query learning, we maintain a momentum-updated *Prototype Bank*, Specifically $P = \{P_c\}_{c=1}^C$, where $P_c \in \mathbb{R}^d$ denotes the prototype for class $c$. For each training image, we compute a per-image prototype $f_c$ by avgpooling $\ell_2$-normalized pixel embeddings within the ground-truth region of class $c$. The Prototype Bank $P_c$ is updated via exponential moving average:

$$P_c \leftarrow \text{norm}(\alpha P_c + (1-\alpha)f_c), \tag{4}$$

where $\alpha$ controls the update rate. To further reduce the intra-class feature variance, we explicitly enforce consistency between $f_c$ and $P_c$ with

$$\mathcal{L}_{\text{img}} = \frac{1}{|\mathcal{C}_b|} \sum_{c \in \mathcal{C}_b} \|f_c - P_c\|_2^2, \tag{5}$$

where $\mathcal{C}_b$ is the set of classes present in the current batch. This regularization encourages per-image prototypes to be close to global anchors, stabilizing training and enhancing feature consistency.

**Alignment Reward.** Directly optimizing image–prototype regularization loss stabilizes training, but query-based segmentation depends on query–image token alignment and interaction, so optimizing only one does not markedly improve segmentation ability. To enable each query to focus on the most relevant class-specific information while avoiding query collapse, we define an alignment reward. Let the refined queries be denoted as $Q_L \in \mathbb{R}^{K \times d}$. For simplicity, we assume that $Q_L$ has already been normalized, and we denote the resulting queries as $Q = \text{norm}(Q_L)$. Given the momentum Prototype Bank $P = \{P_c\}_{c=1}^C$ (Sec. 3.3), the query–prototype similarity matrix $S \in \mathbb{R}^{K \times C}$ is computed as:

$$S = QP^\top, \quad S_{i,j} = \langle Q_i, P_j \rangle, \tag{6}$$

where $S_{i,j}$ represents the cosine similarity between query $i$ and prototype $P_j$. For each query $i$, we select the top-1 class as the most similar class:

$$c_i = argmax_{j \in \{1,\ldots,C\}} S_{i,j}, \quad r_i = S_{i,c_i}. \tag{7}$$

Here, $c_i$ is the index of the most similar class for query $i$, and $r_i$ is the corresponding similarity score, which serves as our alignment reward.

The inner product $S = QP^\top$ is used to compute the query–prototype similarity, consistent with the core principle of query-based methods, where each query's prediction is based on its similarity to class prototypes. By using the similarity score $r_i$ as the reward, we encourage alignment between each query and its most relevant class prototype, ensuring that all queries are effectively trained.

**Group-Relative Advantage.** To foster more efficient query-image tokens interactions, we introduce the Group-relative Advantage approach, inspired by the group-relative advantage concept in

GRPO (Shao et al., 2024). By comparing each query's performance relative to others in its group, we ensure that queries effectively fuse with their most relevant class prototypes, improving segmentation accuracy. We partition the $K$ queries into $G$ groups $\{\mathcal{G}_g\}_{g=1}^{G}$, where each group $\mathcal{G}_g$ consists of queries that share the same most similar prototype $c_i$, as computed in Sec 3.3.

For each group $\mathcal{G}_g$, we compute a baseline defined by the mean and standard deviation of the rewards $r_i$ for the queries in that group:

$$\mu_g = \frac{1}{|\mathcal{G}_g|} \sum_{i \in \mathcal{G}_g} r_i, \quad \sigma_g = \sqrt{\frac{1}{|\mathcal{G}_g|} \sum_{i \in \mathcal{G}_g} (r_i - \mu_g)^2}. \tag{8}$$

Here, $\mu_g$ is the mean reward, and $\sigma_g$ is the standard deviation, with $\varepsilon$ ensuring numerical stability. Next, we define the group-relative advantage for each query $i \in \mathcal{G}_g$ as:

$$A_i = \frac{r_i - \mu_g}{\sigma_g + \varepsilon}. \tag{9}$$

The advantage $A_i$ measures how much query $i$'s reward deviates from the group's baseline. If $A_i > 0$, query $i$ outperforms its group, indicating successful fusion with the most relevant prototype and deserving a reward. If $A_i < 0$, the query underperforms and should be penalized to encourage improvement. This group-relative advantage motivates queries exceeding the baseline to fuse with the most relevant prototypes and enhance their reasoning ability.

**GRPO-style Clipping with Reference KL Stabilization.** While group-relative advantages provide dense supervision, they can exhibit high variance and occasionally induce overly large updates. To keep query updates conservative and stable, we adopt a GRPO/PPO-style clipped objective together with a KL regularization term to an EMA reference model. We maintain a reference model $\theta_{\mathrm{ref}}$, which is an exponential moving average (EMA) of the current model parameters $\theta$. This reference model serves as a stable guide for the current model by providing a reference distribution for comparison. Given the query–prototype similarity matrices $S_\theta = QP^\top$ and $S_{\mathrm{ref}} = Q_{\mathrm{ref}}P^\top$, we convert them into per-query class distributions via softmax:

$$\pi_\theta(i,j) = \frac{\exp(S_\theta[i,j])}{\sum_{j'} \exp(S_\theta[i,j'])}, \qquad \pi_{\mathrm{ref}}(i,j) = \frac{\exp(S_{\mathrm{ref}}[i,j])}{\sum_{j'} \exp(S_{\mathrm{ref}}[i,j'])}. \tag{10}$$

For each query $i$, we define the importance ratio as:

$$\rho_i = \frac{\pi_\theta(i,c_i)}{\pi_{\mathrm{ref}}(i,c_i)}. \tag{11}$$

Following the principles of Proximal Policy Optimization (PPO) and GRPO, we define our group-relative clipped objective as:

$$\mathcal{L}_{\mathrm{GR}} = -\frac{1}{K} \sum_{g=1}^{G} \sum_{i \in \mathcal{G}_g} \min\left(\rho_i A_i, \ \mathrm{clip}(\rho_i, 1-\epsilon, 1+\epsilon) A_i\right), \tag{12}$$

where the clipping function limits the importance ratio $\rho_i$ from deviating excessively from 1, thereby preventing unstable updates. This mechanism ensures that the model's alignment improves gradually, promoting stable learning. Additionally, we regularize the current model's distribution with respect to the reference distribution using forward KL divergence:

$$D_{KL}[\pi_\theta || \pi_{\mathrm{ref}}] = \frac{1}{K} \sum_{i=1}^{K} \left[ \frac{\pi_{\mathrm{ref}}(i,c_i)}{\pi_\theta(i,c_i)} - \log\left(\frac{\pi_{\mathrm{ref}}(i,c_i)}{\pi_\theta(i,c_i)}\right) - 1 \right]. \tag{13}$$

This KL divergence term prevents abrupt shifts in the model's behavior while allowing for gradual and stable improvements. Finally, the overall GRQA alignment loss is defined as:

$$\mathcal{L}_{\mathrm{GRQA}} = -\frac{1}{K} \sum_{g=1}^{G} \sum_{i \in \mathcal{G}_g} \min\left(\rho_i A_i, \ \mathrm{clip}(\rho_i, 1-\epsilon, 1+\epsilon) A_i\right) + \beta D_{KL}[\pi_\theta || \pi_{\mathrm{ref}}], \tag{14}$$

where $\beta > 0$ is a small constant. The GRQA loss captures the alignment between queries and prototypes, while ensuring stable training through the regularization and clipping mechanisms.

**Overall training objective.** Finally, the total training loss combines the standard segmentation loss, the prototype alignment loss, and the GRQA alignment loss:

$$\mathcal{L}_{\text{total}} = \mathcal{L}_{\text{seg}} + \lambda_{\text{img}} \mathcal{L}_{\text{img}} + \lambda_{\text{grqa}} \mathcal{L}_{\text{GRQA}}, \qquad (15)$$

where $\mathcal{L}_{\text{seg}}$ is the supervised segmentation loss, $\mathcal{L}_{\text{img}}$ aligns per-image prototypes with the bank, and $\mathcal{L}_{\text{GRQA}}$ is the group-relative alignment objective. $\lambda_{\text{img}}$ and $\lambda_{\text{grqa}}$ are trade-off weights.

## 4 EXPERIMENTS

**Datasets.** We evaluate QPrompt-R1 on real and synthetic scene datasets, reporting segmentation accuracy and efficiency. The Cityscapes (Citys) dataset (Cordts et al., 2016) includes 2,975 training and 500 validation images at $2048 \times 1024$. We also use BDD100K (BDD) (Yu et al., 2020) and Mapillary (Map) (Neuhold et al., 2017) as out-of-domain benchmarks, with 1,000 and 2,000 validation images at $1280 \times 720$ and $1902 \times 1080$, respectively. We use GTAV (Richter et al., 2016) with 24,966 labeled frames from an open-world simulator for synthetic data,. We also evaluate on four ACDC splits (Sakaridis et al., 2021) for adverse conditions: *Fog*, *Night*, *Rain*, and *Snow*.

**Evaluation setting.** We use three evaluation protocols: (i) **GTAV→Real**: trained on GTAV, tested on Citys, BDD, and Map; (ii) **Real→Real**: trained on Citys, tested on BDD and Map (iii) **Real→ACDC**: trained on Citys, evaluated on ACDC's adverse-condition splits. (iv) **Clean→ Corruptions**: trained on Citys, evaluated on Cityscapes-C. Segmentation accuracy is measured by **mIoU**, and efficiency by **FPS**. Inference speed is reported on a single NVIDIA RTX 4090 GPU with a batch size of 1. Inference is conducted at a resolution of $512 \times 1024$, with real-time baselines evaluated at their official resolutions.

**Implementation details.** We use DINOv2 (Oquab et al., 2023) as the ViT backbone. To obtain fine-grained details in predictions, we use a two-layer transposed-convolution module. Each layer upsamples the logits by a factor of $\times 2$ producing an overall $\times 4$ upsampling. During training, $\mathcal{L}_{\text{seg}}$ is used for the first two-thirds of epochs to train a base model, which is then initialized for the GRQA phase. In the final third, GRQA training is performed, with EMA updating the reference model. Images are cropped into $512 \times 512$ patches using a sliding window.

**Baselines.** We compare against both domain generalization and real-time segmentation methods. For DG baselines, we include Mask2Former (Cheng et al., 2022) with DinoV2 (Oquab et al., 2023), REIN (Wei et al., 2024), FADA (Bi et al., 2024), SoMA (Yun et al., 2025), and MFuser (Zhang & Robby T., 2025), following their reported training settings and input resolutions. For real-time segmentation, we evaluate Next-ViT (Li et al., 2022), RTFormer (Wang et al., 2022), SeaFormer (Wan et al., 2023), PIDNet-L (Xu et al., 2022), SCTNet-B-Seg100 (Xu et al., 2023), GCNet-L (Yang et al., 2025), and EoMT (Kerssies et al., 2025), representing strong and efficient variants. For (iv) **Clean→ Corruptions** setting, we compare with SegFormer (Xie et al., 2021), FAN (Zhou et al., 2022a), TAPADL (Guo et al., 2023) and REIN. All baselines are tested at their recommended inference resolutions for fair comparison.

### 4.1 QUANTITATIVE RESULTS

As shown in Table 1, 2 our method achieves the best balance between accuracy and efficiency, consistently delivering real-time inference at 54 FPS while maintaining strong segmentation accuracy.

**GTAV source (GTAV → Real).** On GTAV-to-real adaptation, our method reaches 64.1 mIoU, surpassing the best real-time baseline (EoMT) by 3.1 mIoU. Its accuracy is comparable to advanced DGSS methods such as REIN, yet our model runs over $\times 5$ faster, ensuring real-time applicability.

**Cityscapes source (Real → Real).** Across real-world datasets, our method achieves 67.8 mIoU, exceeding the best real-time baseline by 1.7 mIoU. Performance is competitive with strong DGSS models such as M2F, while maintaining real-time speed for a superior efficiency–accuracy trade-off.

**Cityscapes source (Real → ACDC).** Under adverse weather conditions, ours attains 69.4 mIoU, improving over EoMT by 3.0 mIoU and showing greater robustness in challenging scenarios. Accuracy approaches leading DGSS methods, while running faster for practical deployment.

**Cityscapes source (Clean → Corruptions).** Across all corruptions, our method achieves the best robustness with 69.8 mIoU, showing clear gains especially under Noise and Blur. It also runs at 54 FPS, offering strong resilience while remaining suitable for real-time use.

Table 1: Comparison between domain generalization semantic segmentation (DGSS) and real-time semantic segmentation (RTSS) methods on GTAV→Real, Real→Real, and Real→ACDC benchmarks. "M2F" denotes Mask2Former, "Seaf" denotes SeaFormer, "Next" denotes Next-ViT, "RTF" denotes RTFormer; "*" indicates our re-implementation using official source code with default settings. All results are reported in mIoU (%) and inference speed in FPS.

| | Method | Refer | GTAV → Real | | | | Real → Real | | | Real → ACDC | | | | | FPS |
|---|---|---|---|---|---|---|---|---|---|---|---|---|---|---|---|
| | | | Citys | BDD | Map | Avg | BDD | Map | Avg | Fog | Night | Rain | Snow | Avg | |
| **DGSS** | M2F* | CVPR22 | 63.7 | 57.4 | 64.2 | 61.7 | 63.7 | 70.4 | 67.1 | 78.4 | 51.9 | 70.5 | 68.9 | 67.4 | 11 |
| | REIN | CVPR24 | 66.4 | 60.4 | 66.1 | 64.3 | 65.0 | 72.3 | 68.7 | 79.5 | 55.9 | 72.5 | 70.6 | 69.6 | 10 |
| | FADA | NeurIPS24 | 68.2 | 61.9 | 68.1 | 66.1 | 65.1 | 75.8 | 70.5 | 80.2 | 57.4 | 75.0 | 73.5 | 71.5 | 8 |
| | SoMA | CVPR25 | 71.8 | 61.3 | 71.6 | 68.2 | 67.0 | 76.5 | 71.8 | 74.7 | 61.7 | 77.8 | 77.3 | 74.4 | 10 |
| | MFuser | CVPR25 | 70.2 | 63.1 | 71.3 | 68.2 | 65.8 | 77.9 | 71.8 | 82.3 | 57.9 | 78.6 | 74.9 | 73.5 | 3 |
| **RTSS** | Next* | Arxiv22 | 50.1 | 30.4 | 40.2 | 40.2 | 52.8 | 60.9 | 56.9 | 71.1 | 20.1 | 54.3 | 49.2 | 51.1 | 57 |
| | RTF* | NeurIPS22 | 45.3 | 26.2 | 38.6 | 36.7 | 43.2 | 56.3 | 49.8 | 69.4 | 16.4 | 49.1 | 43.3 | 44.6 | 94 |
| | Seaf* | ICLR23 | 46.9 | 27.4 | 33.1 | 35.8 | 40.4 | 51.7 | 46.1 | 65.8 | 17.2 | 47.7 | 40.5 | 42.8 | 70 |
| | PIDNet* | CVPR23 | 45.7 | 28.1 | 35.9 | 36.6 | 43.4 | 54.5 | 48.9 | 66.9 | 15.2 | 48.7 | 48.1 | 44.7 | 46 |
| | SCTNet* | AAAI24 | 43.3 | 23.7 | 39.0 | 35.3 | 34.1 | 51.1 | 42.6 | 59.6 | 16.0 | 44.8 | 37.5 | 39.5 | 131 |
| | GCNet* | CVPR25 | 25.7 | 20.9 | 26.9 | 24.5 | 38.0 | 50.8 | 44.4 | 63.0 | 11.1 | 42.4 | 33.1 | 37.4 | 53 |
| | EoMT* | CVPR25 | 62.1 | 57.2 | 63.7 | 61.0 | 62.6 | 69.7 | 66.1 | 77.8 | 52.7 | 69.7 | 65.4 | 66.4 | 52 |
| | Ours | - | **66.1** | **59.0** | **67.1** | **64.1** | **63.8** | **71.7** | **67.8** | **79.5** | **53.1** | **74.2** | **70.6** | **69.4** | 54 |

Table 2: Results on **Cityscapes → Cityscapes-C (level-5)** datasets. In Cityscapes-C, level 5 corresponds to the most severe corruption intensity. "*" indicates our re-implementation using official source code with default setting.

| | Cityscapes → Cityscapes-C (level-5) | | | | | | | | | | | | | | | | | |
|---|---|---|---|---|---|---|---|---|---|---|---|---|---|---|---|---|---|---|
| Method | Blur | | | | Noise | | | | Digital | | | | Weather | | | | Avg | FPS |
| | Motion | Defoc | Glass | Gauss | Gauss | Impul | Shot | Speck | Bright | Contr | Satur | JPEG | Snow | Spatt | Fog | Frost | | |
| SegFormer* | 57.6 | 54.5 | 46.5 | 48.0 | 17.6 | 21.1 | 22.0 | 56.5 | 76.6 | 65.6 | 72.5 | 40.8 | 36.8 | 57.4 | 71.6 | 35.2 | 48.8 | 5 |
| FAN* | 59.4 | 55.1 | 56.3 | 52.5 | 19.3 | 26.3 | 30.1 | 56.1 | 77.4 | 68.0 | 74.3 | 45.7 | 47.9 | 65.6 | 77.3 | 36.5 | 53.0 | 26 |
| TAPADL* | 60.1 | 55.5 | 51.4 | 51.2 | 22.3 | 27.1 | 32.4 | 57.9 | 80.6 | 67.3 | 77.9 | 49.2 | 48.7 | 69.0 | 76.9 | 37.3 | 54.1 | 23 |
| REIN | 68.5 | 71.7 | 69.7 | 68.7 | 6.2 | 23.0 | 13.1 | 63.7 | **81.5** | **78.9** | **80.6** | 68.8 | 63.8 | **73.6** | **79.5** | 47.9 | 60.0 | 10 |
| Ours | **68.7** | **74.5** | **71.1** | **75.0** | **60.4** | **60.9** | **66.7** | **75.1** | 79.4 | 74.8 | 78.4 | **70.1** | **65.5** | 72.5 | 76.0 | **48.0** | **69.8** | **54** |

## 4.2 Qualitative Results

**Segmentation Results.** Exemplar segmentation results are presented in Fig. 3, comparing the performance across the *GTAV → Citys*, *BDD*, and *Map* settings. Our method achieves better pixel-wise predictions than real-time methods, including PIDNet, GCNet, SCTNet , and EoMT.

**Impact of GRQA on Queries.** We train on GTAV and compare two setups: SFT-base (without GRQA post-training) and GRQA post-training. First, we compute the average similarity between queries and class embeddings on Citys, BDD, and Map test sets. As shown in Fig. 4, GRQA increases the similarity between each query and its most relevant class embedding, enhancing query utilization. We also examine query activation rates, defined as queries not assigned to the background class. Across the dataset, a higher number of activated queries indicates that more queries participate in segmentation, reflecting higher query utilization. As shown in Fig. 5, activated queries increased by 45%, 48%, and 52% under GRQA, demonstrating improved query utilization and stronger segmentation of relevant objects.

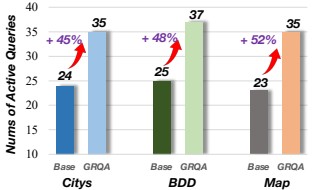

Figure 5: GRQA boosts query activation for segmentation.

## 4.3 Scalability of GRQA

**Plug-and-Play Gains for State-of-the-Art DGSS.** To further assess GRQA versatility, we apply GRQA as a plug-and-play training strategy to two state-of-the-art DGSS methods, REIN and SoMA, under the GTAV→Real setting. As shown in Table 3, GRQA enhances their average performance by +1.2 and +0.6, re-

Table 6: Qprompt with different rewards (trained on GTAV).

| Reward | Trained on GTAV | | | |
|---|---|---|---|---|
| | Citys | BDD | Map | Avg |
| w/o reward | 63.6 | 57.7 | 65.6 | 62.3 |
| w/ DINO-R1 | 64.5 | 58.2 | 66.1 | 62.9 |
| w/ GRQA | 66.1 | 59.0 | 67.1 | 64.1 |

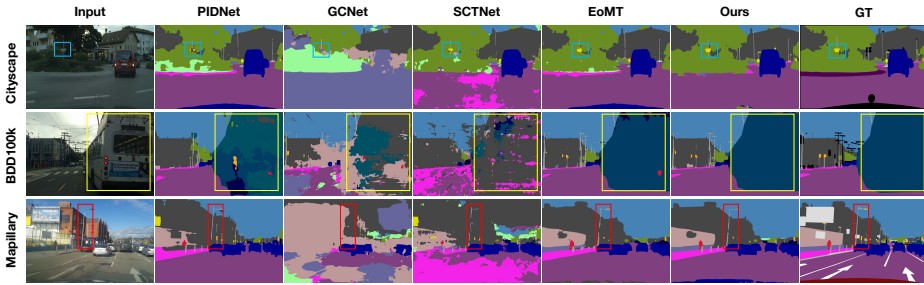

Figure 3: Exemplar segmentation results on *GTAV→Citys*, *BDD*, and *Map*. Compared with real-time baselines, including PIDNet, GCNet, SCTNet, and EoMT, our method delivers noticeably more accurate pixel-wise predictions, highlighting its stronger cross-domain generalization ability.

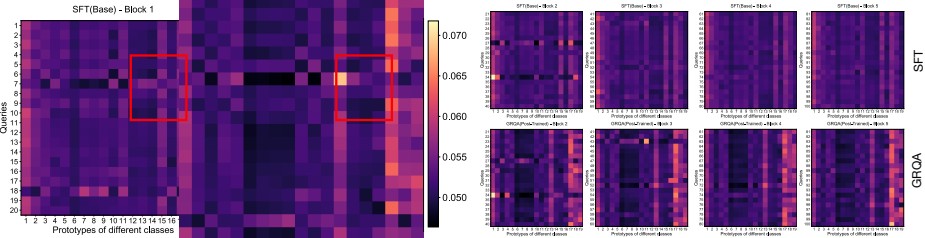

Figure 4: Query–prototype correlations of the Base (SFT) and GRQA models, showing GRQA enhances query–feature fusion. For clarity, $S \in R^{K \times C}$ is split into five blocks.

spectively, without introducing any inference-time overhead. These results indicate that GRQA is a general and effective enhancement for DGSS frameworks, extending its benefits beyond our own model.

## 4.4 ABLATION STUDIES

**Ablation Study on Performance and Efficiency Trade-offs.** We perform an ablation study starting with Mask2Former and progressively removing components, evaluated on GTA5 → Cityscapes. In Table 7, Mask2Former performs well but has low FPS (11). Removing Pixel Decoder boosts FPS but slightly reduces performance. Removing multi-scale has minimal impact on performance and increasing speed. Replacing the Transformer Decoder with an MLP-Head drops performance but

Table 7: Ablation for Performance and Efficiency. Evaluated on setting GTAV *to* Citys

| Method | mIoU | FPS | Param(Infer) |
|---|---|---|---|
| Mask2Former | 63.7 | 11 | 325 M |
| → w/o Pixel Dec | 62.9 | 25 | 320 M |
| → w/o muti-scale | 62.8 | 27 | 320 M |
| → w/o Transformer Dec | 61.3 | 55 | 312 M |
| → QPrompt | 63.6 | 54 | 315 M |
| → QPrompt-R1 | 66.1 | 54 | 315 M |

greatly improves FPS. QPrompt restores performance, maintains high FPS, offering performance-efficiency balance. These results show heavy decoders are a bottleneck, and QPrompt boosts efficiency without compromising performance.

**Variants of QPrompt-R1.** We conduct ablation studies under the *GTAV→Citys+BDD+Map* setting. As shown in Table 4, the baseline (MLP-Head) achieves an average score of 59.9. Introducing QPrompt raises the performance to 62.3, yielding a clear gain of +2.4 and confirming the advantage of our query prompting design over a simple MLP head. Building upon this, image alignment brings a modest improvement of +0.3. The reward mechanism provides a larger boost of +1.1, underscoring its effectiveness in guiding query optimization. Finally, adding KL divergence not only stabilizes training but also brings further gains, achieving the best performance of 64.1. These results highlight the cumulative benefits of GRQA, where

Table 8: Performance of QPrompt-R1 on different VFM backbones.

| Backbone | Method | Citys | Δ |
|---|---|---|---|
| DINOv2-L | MLP-Head | 61.3 | |
| | +Qprompt | 63.6 | +2.3 |
| | +GRQA | 66.1 | +2.5 |
| CLIP-L | MLP-Head | 50.4 | |
| | +Qprompt | 51.7 | +1.3 |
| | +GRQA | 53.2 | +1.5 |
| SAM-H | MLP-Head | 54.7 | |
| | +Qprompt | 55.8 | +1.1 |
| | +GRQA | 57.2 | +1.7 |

Table 3: Performance of GRQA transferred to SOTA DGSS.

| Method | Trained on GTAV | | | | |
|---|---|---|---|---|---|
| | Citys | BDD | Map | Avg | Δ |
| REIN | 66.4 | 60.4 | 66.1 | 64.3 | – |
| +GRQA | 67.4 | 61.0 | 68.1 | 65.5 | +1.2 |
| SoMA | 71.8 | 61.3 | 71.6 | 68.2 | – |
| +GRQA | 72.0 | 62.5 | 71.8 | 68.8 | +0.6 |

Table 4: Roles of individual parts in QPrompt-R1.

| Method | Trained on GTAV | | | | |
|---|---|---|---|---|---|
| | Citys | BDD | Map | Avg | Δ |
| MLP-Head | 61.3 | 55.7 | 62.7 | 59.9 | - |
| QPrompt | 63.6 | 57.7 | 65.6 | 62.3 | +2.4 |
| +img align | 63.9 | 58.2 | 65.8 | 62.6 | +0.3 |
| +Reward | 65.8 | 58.6 | 66.8 | 63.7 | +1.1 |
| +KL | 66.1 | 59.0 | 67.1 | 64.1 | +0.4 |

Table 5: Hyperparameter ablation for GRQA.

| $\epsilon$ | $\beta$ | Citys | $\lambda_{\text{img}}$ | $\lambda_{\text{grqa}}$ | Citys |
|---|---|---|---|---|---|
| 0.05 | 0.001 | 65.6 | 1 | 1 | 64.0 |
| 0.1 | 0.001 | 66.1 | 5 | 1 | 64.5 |
| 0.15 | 0.001 | 64.7 | 10 | 1 | 65.7 |
| 0.1 | 0.01 | 65.5 | 10 | 5 | 66.1 |
| 0.1 | 0.0001 | 65.9 | 10 | 10 | 65.8 |

each component contributes positively, and the full configuration delivers the strongest and most stable generalization.

**Hyperparameter Ablation for GRQA.** We conduct an ablation study to analyze the impact of hyperparameters on GRQA, as shown in Table 5. On the *Citys* dataset, the optimal configuration is $\epsilon = 0.1$ and $\beta = 0.001$, yielding 66.1 mIoU. The results show that $\epsilon$ has minimal impact, while $\beta$ strongly affects performance, highlighting the importance of KL divergence regularization for stabilizing training and ensuring robust query alignment. Additionally, the best performance is achieved with $\lambda_{\text{img}} = 10$ and $\lambda_{\text{grqa}} = 5$. Increasing $\lambda_{\text{img}}$ consistently improves performance, confirming its key role in stabilizing training. In contrast, $\lambda_{\text{grqa}}$ requires careful tuning, as extreme values cause degradation. This indicates that **GRQA** is more sensitive to $\lambda_{\text{grqa}}$ than $\lambda_{\text{img}}$, underscoring the need for balanced integration of the two loss terms.

**Ablation for different rewards.** To further assess the impact of different rewards on our method, we also validated the DINO-R1 (Pan et al., 2025) reward formulation on QPrompt. Since DINO-R1 is designed for object detection, we made necessary adaptations to the method while maintaining the core reward structure as outlined in the original work. Table 6 results show that applying DINO-R1's reward in QPrompt does lead to some improvements. Our proposed GRQA reward still outperforms DINO-R1 in terms of performance.

**Qprompt-R1 on various VFMs.** To investigate whether our method generalizes across different architectures, we evaluate both Qprompt and GRQA under the GTAV→Citys setting on diverse backbones, including DINOv2-L, CLIP-L(Radford et al., 2021), and SAM-HKirillov et al. (2023). As shown in Table 8, both variants consistently improve over the standard MLP-Head baseline. Specifically, Qprompt yields steady gains of +2.3, +1.3, and +1.1, while GRQA further enhances performance with additional improvements of +2.5, +1.5, and +1.7. These results highlight that our approach strengthens DGSS models from both architectural and training perspectives, delivering robust generalization across heterogeneous backbones.

**Parameter, Performance and Speed.** We demonstrate the efficiency and speed benefits of our solution in Table 9. QPrompt achieves a balance between domain generalization, inference speed, and parameter efficiency. It reduces parameters and improves speed while maintaining competitive performance compared to models like Mask2Former, REIN, and FADA. For QPrompt-R1, despite requiring an additional reference model during training, its inference-time parameters remain the same as QPrompt (315M), ensuring efficient speed and count.

Table 9: Performance of QPrompt-R1 on different VFM backbones.

| Method | mIoU | FPS | Total Param |
|---|---|---|---|
| Next-ViT | 50.1 | 57 | 62M |
| Mask2Former | 63.7 | 11 | 325M |
| REIN | 66.4 | 10 | 328M |
| FADA | 68.2 | 8 | 338M |
| QPrompt | 63.6 | 54 | 315M |
| QPrompt-R1 | 66.1 | 54 | 315M |

## 5 CONCLUSION

We introduced QPrompt-R1, a method that simultaneously achieves real-time efficiency and robustness to distribution shifts. Through QPrompt, we inject learnable queries only at the final transformer block, enabling efficient query–image alignment with minimal overhead. Furthermore, our Group-Relative Query Alignment (GRQA) enhances cross-domain robustness via cooperative query supervision—adding no inference cost and integrating seamlessly with existing DGSS models. QPrompt-R1 achieves 54 FPS while maintaining strong cross-domain performance, establishing a new speed–accuracy frontier for semantic segmentation in autonomous driving and robotics.

## ACKNOWLEDGMENTS

This work is supported in part by the National Natural Science Foundation of China (62192783, 62276128, 62406140), Young Elite Scientists Sponsorship Program by China Association for Science and Technology (2023QNRC001), the Key Research and Development Program of Jiangsu Province under Grant (BE2023019) and Jiangsu Natural Science Foundation under Grant (BK20221441, BK20241200). The authors would like to thank Huawei Ascend Cloud Ecological Development Project for the support of Ascend 910 processors.

## ETHICS STATEMENT

All authors have read and agree to abide by the ICLR Code of Ethics. This work does not involve interventions with human participants or personally identifiable information. We use only publicly available datasets under their original licenses and follow the terms of use. Potential risks and our mitigations are summarized below:

- **Privacy & Security.** We do not collect or release any personal data. When showing qualitative examples, all images/videos are from public datasets; any sensitive content is filtered.
- **Bias & Fairness.** We report results on multiple benchmarks and provide detailed settings to facilitate external auditing. We acknowledge possible dataset biases and encourage follow-up evaluation on broader demographics and domains.
- **Dual Use / Misuse.** The method could be misused to enable undesired large-scale labeling or surveillance. To reduce misuse, we release only research artifacts (code/configs) and exclude any tools for scraping or re-identifying individuals.
- **Legal Compliance.** We comply with licenses of all third-party assets (code, models, and datasets) and cite their sources. Any additional third-party terms are respected.
- **Research Integrity.** We document preprocessing, training recipes, and evaluation protocols; random seeds and hyperparameters are provided to enable reproducibility.

Where applicable, institutional review information is withheld for double-blind review and can be provided after acceptance.

## REPRODUCIBILITY STATEMENT

We include training and evaluation details in the main paper and Appendix. Concretely: (i) all hyperparameters, optimization settings, and compute budgets; (ii) full data preprocessing and splits; (iii) code structure with scripts to reproduce the main tables and figures; (iv) checkpoints and logs for the primary models.

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

# A  THE USE OF LLMS

We used ChatGPT-4o to polish our manuscript, using the following prompt:

```
I want you to act as an expert in scientific writing.  I will
provide you with some paragraphs in English and your task is
to improve the spelling, grammar, clarity, conciseness, and
overall readability of the text provided, while breaking down long
sentences, reducing repetition and increasing logic.  You should use
artificial intelligence tools, such as natural language processing,
rhetorical knowledge, and your expertise in effective scientific
writing techniques to reply.  Provide the output as a table in
a readable mode.  The first column is the original sentence, the
second column is the sentence after editing, and the third column
provides explanation of your edits and reasons.  Please edit the
following text in a scientific tone:
```

# B  APPENDIX

## B.1  ABLATION FOR BOUNDARY ACCURACY

To evaluate boundary prediction accuracy, we tested our model under the GTA5 → Cityscapes setting. We compared QPrompt-R1 with the RTSS methods PIDNet, Next-ViT, and the DGSS methods Mask2Former and REIN. .The 1px B-mIoU and 3px B-mIoU metrics measure boundary mIoU at 1 and 3 pixels from the ground-truth boundaries, respectively. As shown in Table 10, our model performs similarly to Mask2Former and REIN, but without multi-scale features or pixel encoders, instead using a lightweight transposed-conv upsampler. This results in speed improvement with higher FPS. These demonstrate that our approach effectively balances efficiency and boundary accuracy, with no substantial loss in segmentation quality.

Table 10: Performance about Boundary accuracy.

| Method | 1px B-mIoU | 3px B-mIoU | mIoU | FPS |
|---|---|---|---|---|
| PIDNet | 25.4 | 29.3 | 45.7 | 46 |
| Next-ViT | 17.1 | 31.6 | 50.1 | 57 |
| Mask2Former | 43.5 | 45.8 | 63.7 | 11 |
| REIN | 44.1 | 46.5 | 66.4 | 10 |
| QPrompt-R1 | 42.7 | 45.3 | 66.1 | 54 |

## B.2  GRQA APPLY TO GENERAL SEGMENTATION

Since the GRQA algorithm leverages mutual supervision between queries for optimization, it can be applied to improve query-based models for general semantic segmentation, not just domain-generalized task. We conduct additional in-domain testing on the Citys→Citys setting. As shown in Table 11, we found that GRQA still provides performance improvement in general semantic segmentation.

## B.3  PERFORMANCE COMPARISON WITH PROMPTED SAM MODELS

We also compared our model with prompt-based methods, specifically FastSAM (Zhang et al., 2023) and Grounded SAM(Ren et al., 2024). Since SAM produces class-agnostic masks, we tested them under the open-vocabulary semantic segmentation setting, where we used text prompts that

Table 11: Performance comparison **general semantic segmentation** (Citys→Citys)

| Method | Citys→Citys (mIoU) |
|---|---|
| QPrompt | 79.2 |
| +GRQA | 80.4 |
| Mask2Former | 82.4 |
| +GRQA | 83.1 |

include class labels to make the predicted masks class-specific. The experiments were conducted on Cityscapes, and as shown in Table 12, our model outperforms both FastSAM and Grounded SAM, achieving significantly higher mIoU and FPS. Additionally, we observed that the speed bottleneck of FastSAM lies in mask classification, rather than in the "everything" mode used for class-agnostic mask prediction.

Table 12: Performance Comparison with Prompted SAM Models.

| Method | mIoU (Citys) | FPS |
|---|---|---|
| FastSAM | 32.6 | < 1 |
| Grounded SAM | 36.7 | < 1 |
| Ours | 66.1 | 54 |

