# OpenReview forum: "QPrompt-R1: Real-Time Reasoning for Domain-Generalized Semantic Segmentation via Group-Relative Query Alignment"
_ICLR.cc/2026/Conference — ICLR 2026 Poster_

### Official Review · Reviewer_dtcA · 2025-10-19

**Soundness:** 2
**Presentation:** 3
**Contribution:** 2
**Rating:** 4
**Confidence:** 5

**Summary:**

This work proposed a unified framework to address real-time segmentation and domain generalized segmentation, which is relevant to real-world applications. The proposed is a plug-and-play solution, which can benefit existing methods without introducing latency overhead. Experiments and ablation studies demonstrate the effectiveness of the proposed method.

**Strengths:**

- This paper provided insights to address real-time segmentation and domain-generalized segmentation in a single framework.
- The proposed GRQA is a plug-and-play module, which can enhance existing real-time segmentation models.

**Weaknesses:**

- As the proposed GRQA is a plug-and-play module, it is also necessary to evaluate atop high-quality, non-real-time segmentation models to verify whether it is consistently beneficial.
- The related work only reviewed CNN-based real-time segmentation methods. More recent state-of-the-art real-time methods like SeaFormer, SeaFormer++ RTFormer, and Next-ViT should be reviewed and compared in the experiments. Moreover, the domain generalization performance under different corruptions should be evaluated on Cityscapes-C and relevant benchmarks and compared against existing methods like SegFormer, FAN, Robustifying Token Attention for Vision Transformers, and Trans4Trans.

**Questions:**

- Would you consider making the source code publicly available to foster future research in this line?
- As this work claimed to address real-time efficiency and domain generalization in a single framework, would you better justify the benefit of your solution for reducing the parameters of the model and improving the inference speed? This is critical and should be discussed in detail.

---

> ### Author Response · Authors · 2025-11-28
> **Author Responses to Reviewer dtcA (1/3 part)**
>
> ---
>
> We would like to sincerely thank you for your efforts and valuable comments to improve our work!
>
> Below we address your concerns.
>
> ---
>
> **Q1: GRQA on High-Quality, Non-Real-Time Models.**
>
> Thank you for your thoughtful and constructive suggestion.
>
> We fully understand your concern regarding the verification of the versatility of the **GRQA** module on high-quality (SOTA), non-real-time segmentation models. We would like to kindly point out that this concern has already been addressed and validated in a specific section of our manuscript.
>
> In our study, we evaluate the performance of GRQA in the section titled “**Plug-and-Play Gains for State-of-the-Art DGSS**,” where we apply **GRQA** to two well-established state-of-the-art (SOTA) domain-generalized semantic segmentation (DGSS) methods: **REIN** and **SoMA**. **Both REIN and SoMA are high-performance models in the DGSS field, and they are non-real-time methods, primarily designed to maximize segmentation accuracy rather than inference speed.**
>
> |           | **Citys** | **BDD**  | **Map**  | **Avg**  |          |
> | --------- | --------- | -------- | -------- | -------- | -------- |
> | REIN      | 66.4      | 60.4     | 66.1     | 64.3     |          |
> | **+GRQA** | **67.4**  | **61.0** | **68.1** | **65.5** | **+1.2** |
> | SoMA      | 71.8      | 61.3     | 71.6     | 68.2     |          |
> | **+GRQA** | **72.0**  | **62.5** | **71.8** | **68.8** | **+0.6** |
>
> As shown in **Table 1**, we apply **GRQA** as a plug-and-play training strategy to **REIN** and **SoMA**, evaluating them under the **GTAV → Real** setting. The results indicate that **GRQA** improves the performance of both models without introducing any inference-time overhead, thus highlighting its general applicability to high-quality models.
>
> ---

---

> ### Author Response · Authors · 2025-11-28
> **Author Responses to Reviewer dtcA (2/3 part)**
>
> **Q2: Comparison with more baseline and Domain Generalization Performance on Cityscapes-C.**
>
> Thanks for your insightful comment.
>
> **1.** **Comparison with more baseline**: Thank you for your suggestion to compare our method against more recent state-of-the-art real-time segmentation models such as **SeaFormer**, **RTFormer**, and **Next-ViT**. We appreciate the importance of including these recent methods for a more comprehensive evaluation. We have updated the manuscript to incorporate this comparison.
>
> We conducted additional experiments to compare our method with the real-time segmentation methods you mentioned. The results are presented in Table 2 (the full table is provided in the manuscript tab 1), where we compare QPrompt-R1 against several recent methods, including Mask2Former, Next-ViT, RTFormer, and SeaFormer, on the GTAV → Real, Real → Real, and Real → ACDC benchmarks.
>
> As shown in the Table 2, **QPrompt-R1 performs competitively in terms of mIoU, while maintaining real-time inference speed, demonstrating the effectiveness of our approach in real-time segmentation tasks.**
>
> | Method       | GTAV → Real (mIoU) | Real → Real (mIoU) | Real → ACDC (mIoU) | FPS  |
> | ------------ | ------------------ | ------------------ | ------------------ | ---- |
> | Mask2Former  | 61.7               | 67.1               | 67.4               | 11   |
> | Next-ViT[1]  | 40.2               | 56.9               | 51.1               | 57   |
> | RTFormer[2]  | 36.7               | 49.8               | 44.6               | 94   |
> | SeaFormer[3] | 35.8               | 46.1               | 42.8               | 70   |
> | QPrompt-R1   | 64.1               | 67.8               | 69.4               | 54   |
>
> **2**.**Domain Generalization Performance on Cityscapes-C[7] (level-5):** We appreciate your suggestion to evaluate domain generalization performance under different corruptions on **Cityscapes-C**.
>
> As requested, we have conducted experiments on **Cityscapes-C (level 5)**, which represents the most severe corruption intensity, and compared our method (QPrompt-R1) with **SegFormer**, **FAN**, **TAPADL**, and **REIN** across various corruption types. The results are summarized in Table 3.
>
> As shown in the table, **QPrompt-R1 outperforms all other methods across multiple corruption types, achieving the highest average mIoU of 69.8** and maintaining a competitive FPS of 54. Notably, QPrompt-R1 demonstrates significant improvements in Blur and Noise categories compared to existing methods, which highlights its effectiveness in handling severe corruptions.
>
> | Corruption     | SegFormer[4] | FAN[5] | TAPADL[6] | REIN     | QPrompt-R1 |
> | -------------- | ------------ | ------ | --------- | -------- | ---------- |
> | Blur Motion    | 57.6         | 59.4   | 60.1      | 68.5     | **68.7**   |
> | Blur Defoc     | 54.5         | 55.1   | 55.5      | 71.7     | **74.5**   |
> | Blur Glass     | 46.5         | 56.3   | 51.4      | 69.7     | **71.1**   |
> | Blur Gauss     | 48.0         | 52.5   | 51.2      | 68.7     | **75.0**   |
> | Noise Gauss    | 17.6         | 19.3   | 22.3      | 6.2      | **60.4**   |
> | Noise Impul    | 21.1         | 26.3   | 27.1      | 23.0     | **60.9**   |
> | Noise Shot     | 22.0         | 30.1   | 32.4      | 13.1     | **66.7**   |
> | Noise Speck    | 56.5         | 56.1   | 57.9      | 63.7     | **75.1**   |
> | Digital Bright | 76.6         | 77.4   | 80.6      | **81.5** | 79.4       |
> | Digital Contr  | 65.6         | 68.0   | 67.3      | **78.9** | 74.8       |
> | Digital Satur  | 72.5         | 74.3   | 77.9      | **80.6** | 78.4       |
> | Digital JPEG   | 40.8         | 45.7   | 49.2      | 68.8     | **70.1**   |
> | Weather Snow   | 36.8         | 47.9   | 48.7      | 63.8     | **65.5**   |
> | Weather Spatt  | 57.4         | 65.6   | 69.0      | **73.6** | 72.5       |
> | Weather Fog    | 71.6         | 77.3   | 76.9      | **79.5** | 76.0       |
> | Weather Frost  | 35.2         | 36.5   | 37.3      | 47.9     | **48.0**   |
> | **Avg**        | 48.8         | 53.0   | 54.1      | 60.0     | **69.8**   |
> | **FPS**        | 5            | 26     | 23        | 10       | **54**     |

---

> ### Author Response · Authors · 2025-11-28
> **Author Responses to Reviewer dtcA (3/3 part)**
>
> **Q3: Source Code Availability.**
>
> Thank you for your suggestion. We plan to release the source code and scripts to foster further research and support reproducibility of our work.
>
> ----
>
>
> **Q4:Justification of Model Efficiency and Inference Speed.**
>
> **A4:** Thank you for your valuable comment.
>
> We understand the importance of clearly justifying the reduction in model parameters and the improvement in inference speed. To address this, we have provided a detailed comparison of the **parameter count** and **inference speed (FPS)** of **QPrompt**, **QPrompt-R1**, and several baseline models, with segmentation performance evaluated on the GTAV$\to$Citys setting. The table 5 below summarizes the results:
>
> | **Method**  | **mIoU (GTAV → Citys)** | **FPS** | **Total Params**         |
> | ----------- | ----------------------- | ------- | ------------------------ |
> | Next-ViT    | 50.1                    | 57      | 62M                      |
> | Mask2Former | 63.7                    | 11      | 325M                     |
> | REIN        | 66.4                    | 10      | 328M                     |
> | FADA        | 68.2                    | 8       | 338M                     |
> | QPrompt     | 63.6                    | 54      | 315M                     |
> | QPrompt-R1  | 66.1                    | 54      | 315M |
>
> From Table 5, it is evident that **QPrompt strikes a balance between domain generalization, inference speed, and parameter efficiency.** Specifically, QPrompt reduces the number of parameters and improves inference speed, while maintaining competitive performance compared to models with heavier decoder heads, such as Mask2Former, REIN, and FADA.
>
> For QPrompt-R1, although the model requires storing an additional **frozen reference model** during the **GRQA training stage**, it is important to note that this **frozen reference model is not used during inference**. As such, QPrompt-R1 has the same inference-time parameters as QPrompt (315M), ensuring that the inference speed and parameter count remain efficient.
>
> In summary, QPrompt reduces the model's parameter count while improving inference speed, offering a more efficient solution without compromising performance. QPrompt-R1 incurs additional parameters during training but maintains the same efficiency during inference, further highlighting the model's practicality for real-time applications.
>
> ----
>
> **References:**
>
> [1] Li J, Xia X, Li W, et al. Next-vit: Next generation vision transformer for efficient deployment in realistic industrial scenarios. arXiv 2022.
>
> [2] Wang J, Gou C, Wu Q, et al. RTFormer: Efficient design for real-time semantic segmentation with transformer. NeurIPS 2022.
>
> [3] Wan Q, Huang Z, Lu J, et al. Seaformer: Squeeze-enhanced axial transformer for mobile semantic segmentation. ICLR 2023.
>
> [4] Xie E, Wang W, Yu Z, et al. SegFormer: Simple and efficient design for semantic segmentation with transformers. NeurIPS 2021.
>
> [5] Zhou D, Yu Z, Xie E, et al. Understanding the robustness in vision transformers. ICML 2022.
>
> [6] Guo Y, Stutz D, Schiele B. Robustifying token attention for vision transformers. ICCV 2023.
>
> [7] Michaelis C, Mitzkus B, Geirhos R, et al. Benchmarking robustness in object detection: Autonomous driving when winter is coming. arXiv 2019.

---

### Official Review · Reviewer_fRjv · 2025-10-30

**Soundness:** 3
**Presentation:** 3
**Contribution:** 3
**Rating:** 6
**Confidence:** 3

**Summary:**

This work proposes QPrompt-R1, a method specifically designed for RT-DGSS (real-time domain generalized semantic segmentation). QPrompt-R1 consists of two components: the QPrompt architecture that ensures real-time performance and GRQA that enhances domain generalization. Based on the VFM backbone, QPrompt injects learnable queries into the last Transformer block to replace the traditional complex decoder. GRQA addresses the insufficient robustness of single-layer query-image interaction. Experiments demonstrate notable performance improvement.

**Strengths:**

- It effectively applies GPRO to segmentation tasks, enhancing segmentation performance without introducing additional inference overhead.
- GRQA can be effectively integrated into current domain generalization semantic segmentation methods to further improve their performance.
- QPrompt-R1 strikes a good balance between domain generalization capability and inference speed.

**Weaknesses:**

- Q-Prompt is similar to EoMT. They both insert queries into the final layer, but the authors discuss no differences.
- In the Evaluation Setting of Section 4.1, the Real→Real scenario should exclude ACDC’s adverse-condition splits.

**Questions:**

- I am curious about whether GRQA is capable of improving the performance of general semantic segmentation. Discussion or even a small experimental analysis would furthur improve this work.
- As the proposed method uses prototye and prototye bank, it is suggested to cite CPT [r1] where category prototype and memory bank are also adopted.

[r1] Quan Tang, et al. Rethinking Feature Reconstruction via Category Prototype in Semantic Segmentation,TIP 2025.

I am more familiar with general semantic segmentation techniques, and follow only mainstream ones in domain transfer. Feel free to correct if there is any misunderstanding.

---

> ### Author Response · Authors · 2025-11-28
> **Author Responses to Reviewer fRjv (1/2 part)**
>
> ---
>
> We would like to sincerely thank you for your efforts and valuable comments to improve our work!
>
> Below we address your concerns.
>
> ---
>
> **Q1: Comparison with EoMT.**
>
> **A1**: Thank you for your constructive comments.
>
> Indeed, both improve inference speed without relying on a heavy head. However, there are important differences in design architecture and training strategy.
>
> * **Simpler, Modular Query Injection:** QPrompt achieves query injection by simply concatenating learnable queries with image tokens before the final layer. It does not rely on any mask attention mechanism, mask predictions from previous layers, or any other auxiliary components. In contrast, EoMT relies on a more intricate structure involving multiple additional mask predictions to provide inputs for its mask attention mechanism. QPrompt introduces minimal modifications to the standard ViT, making it highly modular and seamlessly integrable with the GRQA training strategy.
>
> * **Consistent Training and Inference Process:** EoMT uses a complex, heuristic-driven strategy (e.g., mask-annealing), where auxiliary components (e.g., mask attention) are removed after training to achieve faster inference. This creates a significant train-test process discrepancy. In the Domain Generalization setting, where models already face natural distribution shifts, this artificial gap in the training and testing procedures further destabilizes the model and severely weakens its generalization capability. **QPrompt operates identically during both training and testing.** **Simultaneously, QPrompt's simple, non-heuristic design allows it to integrate seamlessly with GRQA.** While GRQA introduces additional steps during the training phase, these steps are exclusively dedicated to computing the loss and are unrelated to the mask prediction pipeline. Consequently, removing the GRQA component for inference introduces zero divergence in the forward pass, ensuring a consistent and efficient prediction phase.
>
> * **Enabling the Core GRQA Strategy:** QPrompt's primary role is to serve as the essential foundation for our main innovation, the Group-Relative Query Alignment (GRQA) strategy to address RT-DGSS task. The simplicity and inference-consistency of QPrompt directly enable the application of GRQA:
>   * **Stable Reward Mechanism:** QPrompt’s straightforward concatenation mechanism allows us to easily group the queries during training, which is crucial for generating **stable Group Relative Advantages** and the Dense Mutual Supervision among queries. In stark contrast, EoMT utilizes mask annealing, where the query representations are refined by mask attention during training but removed during inference. This disparity leads to **artificially inflated rewards** for queries in the training phase because their optimal representation is unattainable during mask-free inference. Consequently, this inconsistency severely **compromises the accuracy of the group relative reward baseline calculation.**
>   * **Consistency for Generalization:** Because QPrompt does not employ any **heuristic-driven strategies** that alter the model's forward pass, it maintains perfect **training-testing consistency**. This stability is crucial, as introducing a process discrepancy would severely undermine the generalization improvements gained by the GRQA strategy in the domain transfer setting.
>
> We appreciate your valuable feedback. We have updated the manuscript to explicitly include a comparison with EoMT.
>
> ---
>
> **Q2: GRQA in General Semantic Segmentation.**
>
> **A2:** Thank you for your insightful question regarding the applicability of GRQA to general semantic segmentation.
>
> **GRQA** is designed to leverage mutual supervision between queries to optimize query-based models. While it was originally developed for domain-generalized tasks, we hypothesized that it could also improve performance in standard semantic segmentation tasks. To explore this, we conducted additional in-domain testing on the **Cityscapes → Cityscapes** setting.
>
> As shown in **Table 1**, **GRQA provides a performance improvement even in general semantic segmentation,** with QPrompt + GRQA achieving 80.4 mIoU, compared to 79.2 mIoU for QPrompt alone. While the performance gains in this setting are smaller than those observed in domain generalization tasks, GRQA still contributes to enhancing the model’s accuracy. We speculate that domain generalization tasks are inherently more challenging, requiring a greater variety of queries to effectively handle domain shifts, and that mutual supervision is particularly beneficial in such task. This is likely why the performance improvement is more pronounced in domain generalization tasks.
>
> | **Method**  | **Citys → Citys (mIoU)** |
> | ----------- | ------------------------ |
> | QPrompt     | 79.2                     |
> | **+ GRQA**  | **80.4**                 |
> | Mask2Former | 82.4                     |
> | **+ GRQA**  | **83.1**                 |
>
> ---

---

> > ### Author Response · Authors · 2025-11-28
> > **Author Responses to Reviewer fRjv (2/2 part)**
> >
> > **Q3:  Related work about Prototype.**
> >
> > **A3:** Thank you for your suggestion to cite **CPT**, which also uses category prototypes and memory banks in semantic segmentation task. We agree that it provides valuable insights, and we will update our manuscript to include the citation and acknowledge the similarities with our Prototype Bank.
> >
> > ---
> >
> > **W2: Writing typo on in Section 4.1.**
> >
> > **A4:** Thank you for your helpful suggestion. We apologize for the oversight. The **Real→Real** scenario should indeed exclude ACDC's adverse-condition splits. We have already corrected this typo and will conduct further proofreading to improve the overall quality of the manuscript.
> >
> > **"(ii) Real→Real: trained on Citys, tested on BDD, Map, and ACDC’s adverse-condition splits"**
> >
> > to:
> >
> > **"(ii) Real→Real: trained on Citys, tested on BDD and Map"**.

---

### Official Review · Reviewer_9SAk · 2025-10-31

**Soundness:** 3
**Presentation:** 2
**Contribution:** 3
**Rating:** 6
**Confidence:** 4

**Summary:**

This paper investigates the problem of real-time domain-generalizing semantic segmentation. The authors propose to depart from standard, heavy decoders which are used for this dense classification task on top of vision foundation model encoders, and replace these decoders with a single, light-weight and efficient transformer decoder layer. To enrich the mask-level queries which are involved in this decoder layer, they propose a novel group-relative query alignment module which produces a reinforcement-learning-type reward, optimized in a standard supervised fashion. The authors validate their model on several central domain generalization benchmarks, achieving very high efficiency while staying competitive in terms of segmentation performance to state-of-the-art generalizable segmentation methods.

**Strengths:**

1. The examined problem, albeit constituting a simple combination of two existing ones - i.e. real-time semantic segmentation and domain-generalizing semantic segmentation, is underexplored and yet very relevant for practical real-world applications of Computer Vision, including safety-critical settings that involve uncontrolled and unconstrained operational design domains and at the same time fast inference, such as autonomous vehicles and field robotics.

2. The idea to depart from complex, MaskFormer-style decoders which are the current de-facto standard for semantic segmentation and to leverage the features which are already extracted by Vision Foundation Models in their image tokens is interesting and promising, even though the authors have not provided experimental evidence that the bottleneck of recent segmentation models in terms of latency are indeed these decoders.

3. The proposed Group-Relative Query Alignment is a novel technique in the context of semantic segmentation, with a strong and rigorous theoretical formulation and a demonstrated clear performance benefit for diverse VFM-based segmentation networks (cf. Table 6). This novel module is relevant both in the context of real-time semantic segmentation and beyond that, as it delivers performance gains for heavier, more highly performant yet less efficient networks for domain-generalizing semantic segmentation (cf. Table 2).

**Weaknesses:**

1. Missing evidence for motivation on efficiency-level design decisions. The authors state in the very beginning of the paper in the Abstract that "the bottleneck in Domain Generalization is the prediction head, not the backbone". However, this statement is not supported by experimental evidence, e.g. by benchmarking a baseline network of similar architecture to the one proposed which however does include standard pixel and transformer decoders - a la Mask(2)Former - in terms of inference computation in order to break down the inference latency / FLOPS into backbone-level and head-/decoder-level portions. Combined with the fact that in the main experimental comparison (Table 1) the proposed model trails the performance of state-of-the-art, heavier methods, most of which feature the above decoders, the precise quality of the tradeoff between performance and efficiency is not clear.

2. Omission of learned upsampling to handle fine-grained details in segmentations. The utilized VFMs, such as ViT, rely on patchification to tokenize the input image, which heavily reduces the resolution of their output image-token-based features. Yet, the proposed method operates directly on these heavily downsampled features to produce the final segmentation map. That is, the method does not involve any learned upsampling, exactly of the type that is performed in the pixel decoder module of standard mask-based segmentation architectures such as Mask(2)Former which is omitted in the presented architecture. Thus, the question arises on whether the resulting predictions are deteriorated in terms of preservation of fine-grained details in the ground-truth semantic boundaries.

3. Issues with the presentation of the method. The overall presentation of the method is of fair quality, however, there are a few unclear points. The symbol $K$ is never properly defined and it is not clear whether it denotes the number of semantic classes in the set, as is usually the case in the notation of semantic segmentation papers. This potential relation is reinforced by the fact that in Fig. 4, the number of queries $K$ is 20, which differs from the number of class prototypes (19) only by 1. In Eq. (7) and (8), the parameter $\epsilon$ appears twice; only a single addition of a positive constant is nonetheless required to avoid an ill-defined zero-valued denominator in (8). The core motivation behind the full GRQA objective introduced in p. 6 is not straightforward to me. Some background on this type of reinforcement learning could be useful for readers who are not familiar with the topic. In this context, L. 268-271 are unstructured and hard to parse.

**Questions:**

1. Can the authors provide an "end-to-end" ablation starting from a baseline, heavy network with pixel and transformer decoder heads on top of the VFM backbone, and then first remove these heads and subsequently add the proposed components one by one, finally arriving at the fully-fledged proposed network? Such an ablation should include both performance figures and efficiency figures, importantly latency and amount of computation, and potentially also parameter count and total space complexity at inference. This would greatly help in supporting the motivation that efficiency can be substantially improved by removing the decoder heads.

2. Can the authors provide clarifications on the GRQA part of their method? (cf. Weaknesses)

3. Can the authors provide a comparison in terms of boundary accuracy of the competing segmentation methods against their method? This would help to understand whether the proposed efficiency improvement preserves fine-grained details in the segmentation output despite working with lower-resolution features.

---

> ### Author Response · Authors · 2025-11-28
> **Author Responses to Reviewer 9SAk (1/3 part)**
>
> ---
>
> We would like to sincerely thank you for your efforts and valuable comments to improve our work!
>
> Below we address your concerns.
>
> ---
>
> **Q1:End-to-End Ablation Study on Performance and Efficiency.**
>
> We thank for your insightful suggestion to provide a detailed "end-to-end" ablation study starting from a baseline network and progressively removing components.
>
> We conducted a comprehensive ablation study to evaluate the performance and efficiency trade-offs of our proposed QPrompt architecture, **starting from a baseline Mask2Former** network and systematically removing components on the setting GTAV $\to$ Citys. The ablation results, summarized in the following table, clearly demonstrate the efficiency gains achieved by progressively removing the decoder heads:
>
> | **Method**                                                   | **mIoU** | **$\Delta_{\text{mIoU}}$** | **FPS** | **$\Delta_{\text{FPS}}$** | **Param (inference)** |
> | ------------------------------------------------------------ | -------- | -------------------------- | ------- | ------------------------- | --------------------- |
> | Mask2Former                                                  | 63.7     | -                          | 11      | -                         | 325M                  |
> | &nbsp;&nbsp;&nbsp;&nbsp; $\rightarrow$ w/o Pixel Decoder     | 62.9     | -0.8                       | 25      | +14                       | 320M                  |
> | &nbsp;&nbsp;&nbsp;&nbsp;&nbsp;&nbsp;&nbsp; $\rightarrow$ w/o Multi-scale | 62.8     | -0.1                       | 27      | +2                        | 320M                  |
> | &nbsp;&nbsp;&nbsp;&nbsp;&nbsp;&nbsp;&nbsp;&nbsp;&nbsp;&nbsp;&nbsp; $\rightarrow$ w/o Transformer Decoder | 61.3     | -1.5                       | 55      | +28                       | 312M                  |
> | &nbsp;&nbsp;&nbsp;&nbsp;&nbsp;&nbsp;&nbsp;&nbsp;&nbsp;&nbsp;&nbsp;&nbsp;&nbsp;&nbsp;&nbsp; $\rightarrow$ QPrompt | 63.6     | +2.3                       | 54      | -1                        | 315M                  |
> | &nbsp;&nbsp;&nbsp;&nbsp;&nbsp;&nbsp;&nbsp;&nbsp;&nbsp;&nbsp;&nbsp;&nbsp;&nbsp;&nbsp;&nbsp;&nbsp;&nbsp;&nbsp;&nbsp;&nbsp;&nbsp;&nbsp;&nbsp; $\rightarrow$ QPrompt-R1 (+ GRQA) | 66.1     | +2.5                       | 54      | 0                         | 315M                  |
>
> As shown in the table, we begin with Mask2Former, which includes both a pixel encoder and a transformer decoder. While it performs well in terms of mIoU (63.7), it suffers from low FPS (11). When the pixel decoder is removed, we see a slight drop in performance (62.9 mIoU), but a significant boost in FPS (from 11 to 25). Removing the multi-scale component results in a minimal performance drop (62.8 mIoU), with a further increase in FPS (27).
>
> Next, when we replace the transformer decoder with an MLP head, there is a noticeable drop in performance (61.3 mIoU), but the FPS increases dramatically (55). **This shows that heavy decoders are indeed a bottleneck in terms of speed.** Introducing **QPrompt** restores performance (63.6 mIoU), maintains a high FPS (54), and offers a good balance between performance and efficiency. Finally, **QPrompt-R1** (which includes both QPrompt and the Group-Relative Query Alignment (GRQA) strategy) achieves the highest performance (66.1 mIoU), while maintaining the same FPS (54), demonstrating that the combination of QPrompt and GRQA not only boosts efficiency but also improves performance.
>
>
>
> ---

---

> > ### Author Response · Authors · 2025-11-28
> > **Author Responses to Reviewer 9SAk (2/3 part)**
> >
> > **Q2: Clarification on Notation, Equations and Motivations.**
> >
> > **A2:** Thank you for your helpful comment.
> >
> > **1. Notation ($K$ vs. $C$).**
> >
> > We apologize for any confusion caused by Figure 4. In our notation, **$K$ represents the number of queries** (set to $K=100$), while **$C$ denotes the number of classes** (set to $C=19$). For visual clarity in the figure, we illustrated the similarity matrix $S \in \mathbb{R}^{K \times C}$ (specifically $100 \times 19$) by partitioning it into five smaller blocks. We have explicitly clarified this relationship between $K$ and $C$ and the visual partitioning technique in the revised manuscript.
> >
> > **2. Equations (7) and (8) Simplification.**
> >
> > Thank you for pointing this out. We have revised Equations (7) and (8) and will conduct further proofreading to improve the overall quality of the manuscript.
> >
> > **3. Clarification of the Motivation Behind GRQA.**
> >
> > We sincerely thank you for your insightful comment. To address your concern, we would like to provide a clearer explanation of the motivation behind the **Group-Relative Query Alignment (GRQA)[1]** objective.
> >
> > **(a) Background on GRPO**: Group-Relative Policy Optimization (GRPO) is a simplified policy-optimization framework that replaces the value function with a **group-relative advantage**. Given a group of outputs, each output’s advantage is computed by subtracting the **group mean reward**:
> > $\hat{A}_i = r_i - \frac{1}{G} \sum^G_j r_j.$ GRPO updates the policy using a PPO-style **clipped ratio objective**:  $\min(\rho_i \hat{A}_i, \text{clip}(\rho_i,1-\epsilon,1+\epsilon)\hat{A}_i),$ and includes a **KL regularization term** that keeps the policy close to a reference model, ensuring conservative and stable updates.
> >
> > The overall objective of GRPO is: $\mathcal{L}_{\text{GRPO}} = \mathbb{E} \left[ \min\left( \rho_i \hat{A}_i , \mathrm{clip}(\rho_i, 1-\epsilon, 1+\epsilon) \hat{A}_i \right) \right] - \beta  KL   $
> >
> > **(b) Motivation for the full GRQA objective**:
> >
> > * **Motivation for Group-Relative Advantages:** Previous query-based segmentation methods typically rely on Hungarian matching, which assigns one query per ground truth. This approach results in **sparse supervision** and inherently limits the robustness of the query set. Inspired by **GRPO**, which computes relative rewards automatically, we aimed to introduce a dynamic, **group-level supervision signal**. By leveraging the relationships among queries within a group, we provide a **dense, self-calibrated mutual supervision signal** that enhances the model’s robustness.
> > * **Motivation for GRPO-style Clipping with Reference KL Stabilization:** While the group-relative advantages offer dense supervision, they can also exhibit high variance, which might **lead to excessively large, unstable updates** due to the dynamic nature of the group baseline. In the original GRPO framework, this issue is mitigated through the use of a PPO-style clipped objective and a KL regularization term to ensure that optimization steps remain both trustworthy and conservative. **To stabilize our updates,** we adopt a similar approach, using the GRPO/PPO-style clipped objective combined with a KL regularization term. This term operates on an reference model, which serves as a stable guide for the current model by providing a reference distribution for comparison. The distance between current and reference model is measured by the per-query class distributions ($\pi_\theta$ and $\pi_{\mathrm{ref}}$).
> >
> > In summary, the full GRQA objective combines **group-relative advantages** for dense supervision with **GRPO-style clipping and reference KL stabilization** for stable updates, to address the limitations of sparse supervision and enhance generalization,.

---

> > > ### Author Response · Authors · 2025-11-28
> > > **Author Responses to Reviewer 9SAk (3/3 part)**
> > >
> > > **Q3: Omission of Learned Upsampling for Fine-Grained Details.**
> > >
> > > Thank you for your constructive suggestion.
> > >
> > > We appreciate your concern regarding the potential loss of fine-grained details in segmentation due to the lack of learned upsampling in our proposed method. We would like to clarify this point and provide further experimental results to address your question.
> > >
> > > **1. Learned Upsampling in Our Method**: Although our method does not use a **pixel encoder module** with learnable upsampling, as seen in traditional mask-based segmentation architectures like Mask2Former, **we do include an upsampling mechanism in our architecture.** Specifically, **we employ two ×2 transposed convolution layers to upsample the output of the VFM,** resulting in a total **upsampling factor of ×4.** This lightweight upsampling mechanism enhances the resolution of the output segmentation map without significantly increasing computational overhead. We apologize for any confusion caused and have updated the manuscript to clarify that our method relies on this transposed convolution upsampler for resolution enhancement, while maintaining efficiency in real-time applications.
> > >
> > > **2. Boundary Accuracy Comparison**:
> > > To address your concern about the preservation of fine-grained details, we conducted boundary accuracy experiments to evaluate how well our method preserves the boundaries of ground-truth segmentations. We compared QPrompt-R1 with the RTSS methods PIDNet, Next-ViT, and the DGSS methods Mask2Former and REIN. Specifically, we used the **1px B-mIoU** and **3px B-mIoU** metrics to assess boundary accuracy at 1 and 3 pixels from the ground-truth boundaries, under the GTA5 $\to$ Citys setting.
> > >
> > > | **Method**      | **1px B-mIoU** | **3px B-mIoU** | **mIoU** | **FPS** |
> > > | --------------- | -------------- | -------------- | -------- | ------- |
> > > | **PIDNet**      | 25.4           | 29.3           | 45.7     | 46      |
> > > | **Next-ViT[2]** | 17.1           | 31.6           | 50.1     | 57      |
> > > | **Mask2Former** | 43.5           | 45.8           | 63.7     | 11      |
> > > | **REIN**        | 44.1           | 46.5           | 66.4     | 10      |
> > > | **QPrompt-R1**  | 42.7           | 45.3           | 66.1     | 54      |
> > >
> > > Our method performs similarly to **Mask2Former** and **REIN**, both of which rely on multi-scale features and pixel encoders. Despite not using these techniques, our model achieves 42.7 for 1px B-mIoU and 45.3 for 3px B-mIoU, with a significant increase in FPS (54). **These results demonstrate that our method successfully balances boundary accuracy and efficiency,** **without a substantial loss in segmentation quality, even without multi-scale features or pixel encoders.**
> > >
> > >
> > >
> > > **References:**
> > >
> > > [1] Shao Z, Wang P, Zhu Q, et al. Deepseekmath: Pushing the limits of mathematical reasoning in open language models. arXiv 2024.
> > >
> > > [2] Li J, Xia X, Li W, et al. Next-vit: Next generation vision transformer for efficient deployment in realistic industrial scenarios. arXiv 2022.

---

### Official Review · Reviewer_Zb67 · 2025-11-02

**Soundness:** 3
**Presentation:** 2
**Contribution:** 3
**Rating:** 6
**Confidence:** 3

**Summary:**

The paper proposes QPrompt-R1, a method addressing the novel and practical problem of Real-Time Domain-Generalized Semantic Segmentation (RT-DGSS). The core idea is to inject a small set of learnable queries into the final layer of a Vision Foundation Model (VFM) backbone (QPrompt) and enhance their training with a novel Group-Relative Query Alignment (GRQA) objective. This approach aims to retain the domain generalization benefits of query-based methods while achieving real-time speeds by avoiding heavy decoder stacks. The results are impressive, showing a strong balance of speed and robust cross-domain performance.

**Strengths:**

1. Generalizability of GRQA: The demonstration that GRQA is a "plug-and-play" module that can be added to other SOTA DGSS methods to improve their performance is a major strength.
2. Motivation: The paper is well-motivated, identifying a critical and practical problem.

**Weaknesses:**

1. Overclaiming of the "First" RT-DGSS Problem: While this paper is the first to formally name and define the problem, the concept of achieving real-time and robust segmentation is an implicit goal in the field. RTSS papers also evaluate cross-domain performance and DG papers evaluate the computational efficiency, even if it's not their primary focus.
2. Novelty of QPrompt: The idea of injecting tokens/queries is not new. The prototype bank, which is claimed to be a key component of the GRQA strategy, has already been widely applied. The specific application to create a lightweight, single-layer "prompting" mechanism for segmentation might be novel in this context, but the framing should be more precise to acknowledge the inspiration.

**Questions:**

Baseline Comparison: Could you include a comparison with a prompted version of SAM or other fast, promptable foundation models to better contextualize the performance of your QPrompt architecture?

---

> ### Author Response · Authors · 2025-11-28
> **Author Responses to Reviewer Zb67 (Part 1/1)**
>
> ---
>
> We would like to sincerely thank you for your efforts and valuable comments to improve our work!
>
> Below we address your concerns.
>
> ---
>
> **Q1:Overclaiming of the "First" RT-DGSS Problem.**
>
> **A1:** We sincerely appreciate your keen observation and constructive feedback on this point.
>
> In response to your valuable suggestion, we have revised our contribution claim to: **"We highlight Real-Time Domain Generalized Semantic Segmentation (RT-DGSS) as an important and practical research challenge,  addressing both robustness to domain shifts and real-time inference efficiency.".**
>
> Despite the **implicit pursuit** of real-time and domain generalization segmentation in existing research,  **the importance of the Real-Time Domain-Generalized Semantic Segmentation (RT-DGSS) problem has not been given the attention it deserves.** As demonstrated in Table 1(full table can refer to our manuscript in tab 1), previous RTSS or DGSS approaches have struggled to strike a reasonable balance between performance and speed, primarily because they did not explicitly optimize for both aspects concurrently. This oversight has resulted in unsatisfactory trade-offs in the design of methods for RT-DGSS. In contrast, **we emphasize the importance of addressing RT-DGSS as an explicit research challenge.** By focusing on this issue in our design, we have achieved a better balance in RT-DGSS task. **Thus, our contribution lies in highlighting the importance of this challenge and providing a framework that explicitly addresses this balance.** We believe that this explicit focus distinguishes our work and offers a novel perspective in the context of real-time domain-generalized semantic segmentation.
>
> We appreciate your valuable feedback, and we have revised the manuscript accordingly.
>
> | Method      | GTAV → Real (mIoU) | Real → Real (mIoU) | Real → ACDC (mIoU) | FPS  |
> | ----------- | ------------------ | ------------------ | ------------------ | ---- |
> | Mask2Former | 63.7               | 67.4               | 67.4               | 11   |
> | REIN        | 64.3               | 68.7               | 69.6               | 10   |
> | PIDNet      | 36.6               | 48.9               | 44.7               | 46   |
> | SCTNet      | 35.3               | 42.6               | 39.5               | 131  |
> | QPrompt-R1  | 64.1               | 67.8               | 70.6               | 54   |
>
> ---
>
> **Q2: Novelty of QPrompt.**
>
> **A2:** Thanks for your valuable comments.
>
> Our core contribution lies in the development of an efficient and robust semantic segmentation model, along with a plug-and-play GRQA training strategy designed to enhance the model's generalization. The details of the novelty in our method are as follows:
>
> **One of our key innovations is the design of a lightweight, single-layer 'prompting' mechanism integrated into the final Transformer layer.** This design replaces the computation-intensive heads commonly seen in query-based models, while harnessing the capabilities of VFMs to preserve strong domain generalization. It effectively achieves a balance between performance and speed in the RT-DGSS task. This constitutes a central aspect of our contribution.
>
> **Another core innovation is the Group-Relative Query Alignment (GRQA) training strategy.** GRQA incorporates the concept of group-relative rewards from GRPO, leveraging group-relative advantages to jointly optimize all queries. Additionally, it employs GRPO-style clipping combined with reference KL stabilization to ensure training stability. GRQA significantly enhances robustness to domain shifts while maintaining inference-time efficiency.
>
> We have revised the paper to acknowledge the valuable inspiration drawn from previous work.
>
> ---
>
> **Q3:Baseline Comparison with Prompted SAM Models.**
>
> **A3:** Thank you for your valuable suggestion.
>
> We thank for the suggestion to include a comparison with prompt-based methods, specifically **FastSAM** and **Grounding SAM**, to better contextualize the performance of our QPrompt architecture.
>
> We conduct a comparison with **FastSAM[1]** and **Grounded SAM[2]** under the open-vocabulary semantic segmentation setting, where we use class labels as text prompts to make predicted masks class-specific. These experiments are carried out on the Citys. As shown in **Table 1**, **Ours outperforms both FastSAM and Grounding SAM, achieving higher mIoU and FPS.**
>
> We hope these comparisons provide a clearer context for the performance of QPrompt relative to other promptable foundation models.
>
> |                 | mIoU(Citys) | FPS    |
> | --------------- | ----------- | ------ |
> | FastSAM[1]      | 22.6        | <1     |
> | Grounded SAM[2] | 36.7        | <1     |
> | QPrompt-R1      | **66.1**    | **52** |
>
> **References:**
>
> [1] Zhang C, Han D, Qiao Y, et al. Faster segment anything: Towards lightweight sam for mobile applications. arXiv 2023.
>
> [2] Ren T, Liu S, Zeng A, et al. Grounded sam: Assembling open-world models for diverse visual tasks. arXiv 2024.

---

### Author Response · Authors · 2025-11-28
**Author General Responses**

-----

**We would like to sincerely thank all reviewers for your efforts and valuable comments to improve our work!**

------

**We have uploaded a new version of our main paper and supplementary material, revised based on reviewers’ valuable and helpful comments. We highlight the revised parts in blue color for better reference.**

---

**The Core Motivation of our work:**

Our work is motivated by a critical yet practical challenge: achieving both efficiency and robustness in semantic segmentation. Existing methods however, have largely overlooked the joint optimization of these two factors, and thus struggle to satisfy these dual demands simultaneously (Table 1). To address this, we present the framework to jointly optimize these factors through our model architecture, **QPrompt**, and training strategy, **GRQA**. Specifically, we identified that the efficiency bottleneck in high-quality models lies in their heavy segmentation heads (Table 7) and designed QPrompt to mitigate this issue. Furthermore, to enhance domain generalization without incurring additional inference overhead, we propose GRQA, a query optimization-based training strategy. Our final model, **QPrompt-R1**, successfully tackles the Real-Time Domain Generalization Semantic Segmentation (RT-DGSS) task, striking an optimal balance between inference speed and generalization capability.

---

**We are pleased that all reviewers consistently appreciate the novelty and practicality of the problem we proposed, the versatility and effectiveness of our proposed method, and the strong balance achieved between speed and performance.**

**1. Reviewers appreciate the novelty and practical significance of the problem (RT-DGSS):**

- The paper addresses the novel and practical problem of Real-Time Domain-Generalized Semantic Segmentation (RT-DGSS). (Reviewer `Zb67`)
- The paper is well-motivated, identifying a critical and practical problem. (Reviewer `Zb67`)
- This paper investigates the problem of real-time domain-generalizing semantic segmentation, which is relevant to real-world applications. (Reviewer `9SAk`, `dtcA`)
- The approach proposes to depart from standard, heavy decoders, which is a key innovation for achieving real-time speeds. (Reviewer `9SAk`)

**2. Reviewers praise the practical value and inspiration of the proposed method:**

- The proposed QPrompt-R1 method offers an efficient solution for real-time segmentation while maintaining high domain generalization performance, making it highly relevant for practical, safety-critical applications. (Reviewer `fRjv`, `dtcA`)
- The departure from heavy decoders to a lightweight, efficient transformer decoder layer provides a clear practical advantage in real-time segmentation. (Reviewer `9SAk`)

**3.Reviewers commend the transferability of GRQA and the simplicity of QPrompt:**

- The Group-Relative Query Alignment (GRQA) module is praised for its ability to be integrated as a "plug-and-play" solution into existing domain-generalizing segmentation models, offering enhanced performance. (Reviewer `Zb67`, `fRjv`)
- The QPrompt architecture is lauded for its simplicity and effectiveness, injecting learnable queries into the final layer of the Vision Foundation Model (VFM) backbone to achieve real-time segmentation without adding significant overhead. (Reviewer `9SAk`)

**4. Reviewers appreciate the strong and rigorous theoretical formulation:**

- The theoretical formulation of the method is praised for its rigor, with reviewer specifically noting that the method demonstrates clear performance benefits across diverse VFM-based segmentation networks. (Reviewer `9SAk`)

---

> ### Author Response · Authors · 2025-12-03
> **Summary of Revisions**
>
> We thank all reviewers for their careful reading and constructive suggestions.
>
> We have uploaded a revised version of both the main manuscript and the supplementary material, with all updated content highlighted in blue for ease of checking. Below we outline how we have addressed the core concerns from each reviewer and strengthened the work accordingly.
>
> ### **Reviewer-wise Summary of Concerns and Resolutions**
>
> The *`reviewer Zb67 (Rating: 6)`* primarily expresses concerns about **our claim regarding the RT-DGSS problem and the clarity of QPrompt's novelty boundary.** In response, we have revised our claim regarding the RT-DGSS problem to avoid overstatement and have more clearly clarified our contributions, while providing a more precise definition of QPrompt's novelty. Additionally, as suggested, **we expand the baseline comparisons by including prompted SAM models** such as FastSAM and Grounded SAM, demonstrating our method's advantages in both segmentation accuracy and speed.
>
> The *`reviewer 9SAk (Rating: 6)`* raises concerns about **the lack of an end-to-end ablation** to justify the efficiency-level design decisions and the need for a boundary accuracy analysis. In response, we conduct a comprehensive end-to-end ablation study starting with Mask2Former, which includes a heavy decoder head, and progressively remove components. The results demonstrate that the heavy decoder head in Mask2Former is a bottleneck for inference speed, which supports our design decisions. We also include **a boundary accuracy analysis**, showing that our method improves inference speed without compromising boundary precision, effectively balancing efficiency and segmentation quality. Furthermore, We follow the reviewer’s suggestion by improving the presentation of our method, including updating the notation, adding background information on reinforcement learning (RL), and refining the text to better articulate the motivation behind our approach.
>
> The *`reviewer fRjv (Rating: 6)`* expresses concerns about the **applicability of the GRQA method to general semantic segmentation tasks**. In response, we apply GRQA in a general semantic segmentation setting and showed that integrating the GRQA training strategy improves both QPrompt and Mask2Former by 1.2 and 0.7, respectively. The results indicate that GRQA is effective in general semantic segmentation as well and functions as a plug-and-play module. Additionally, following the reviewer’s suggestion, we **clarify the relationship between QPrompt and EoMT**, emphasizing the differences in model architecture, training strategy, and compatibility with GRQA. We also **incorporate the recommended citation** to related work and correct writing errors in the manuscript.
>
> The *`reviewer dtcA (Rating: 4)`*  expresses concerns about **the need for more baseline comparisons**. In response, we compare our method against additional real-time segmentation models mentioned by reviewer, including SeaFormer, RTFormer, and Next-ViT, and demonstrate that our method outperforms these models in domain generalization while maintaining a balanced trade-off between speed and performance. Additionally, we conduct a **parameter and speed analysis**, showing a $5\times$ speed improvement and a 10M reduction in parameters compared to Mask2Former, emphasizing our method’s efficiency and domain generalization advantages. Furthermore, following the reviewer’s suggestion, we **evaluate our method on Cityscapes-C**, demonstrating that it maintains strong generalization under corruption settings while achieving real-time inference speed. **For the applicability to high-quality, non-real-time models,** we have previously demonstrated the effectiveness of GRQA with REIN and SoMA, both of which are high-quality, non-real-time models, in earlier submissions. Our results indicate improvements with GRQA, confirming that it is a plug-and-play solution.

---

> ### Author Response · Authors · 2025-12-03
>
> ### **Detailed Reviewer-wise Summary of Concerns and Resolutions**
>
> 1. We appreciate the constructive suggestions from *Reviewer `Zb67`* . Below, we provide our responses to the concerns and questions raised.
> * **Concern about overclaiming the "First" RT-DGSS Problem:** Following the reviewer’s suggestion, we have revised our claim regarding the RT-DGSS problem to avoid overstatement and more clearly and precisely highlighted our contributions.
> * **Suggestion for clarifying the novelty of QPrompt:** Following the reviewer’s suggestion, we rephrase the novelty of QPrompt to make its innovative boundaries more explicit. We also incorporate more related work and acknowledge their influence on our approach.
> * **Request for evaluations on prompted SAM models:** Following the reviewer’s request, we add baseline comparisons with FastSAM and Grounded SAM. The results show that our method outperforms these prompted SAM models, highlighting the effectiveness and efficiency of our approach.
>
> 2. We thank *Reviewer `9SAk`* for the helpful feedback. Below, we address the main concerns and issues brought up in the review.
> * **Request for End-to-End Ablation Study on Performance and Efficiency:** In response to the reviewer’s request, we conduct an ablation study starting from Mask2Former with a heavy decoder head and progressively remove components. The results showed that Mask2Former’s heavy decoder head is a bottleneck for inference speed, providing insight into our efficiency-level design decisions.
> * **Suggestion for Issues with the Presentation of the Method:** Following the reviewer’s suggestion, we revise the manuscript for clarity, update the notation, add background information on reinforcement learning (RL), and refine the text to better articulate the motivation behind our approach.
> * **Concern about Upsampling Method and Boundary Accuracy in Our Approach:** In response to the reviewer’s concern, we first provide details on our upsampling process, which involves two transposed convolution layers. We then report boundary accuracy comparisons between the baseline and ours. The results demonstrates that our approach improves inference speed without compromising boundary precision.
>
> 3. We are grateful to *Reviewer `fRjv`* for the valuable comments. Below, we respond to the key concerns highlighted in the review.
> * **Request for clarification on the Relationship between Q-Prompt and EoMT:** In response to the reviewer’s request, we further clarify the differences between the two in terms of model architecture, training strategy, and compatibility with GRQA.
> * **Concern about applicability to General Semantic Segmentation Tasks:** Following the reviewer’s suggestion, we conduct an ablation study showing that integrating the GRQA training strategy improves QPrompt and Mask2Former by 1.2 and 0.7, respectively. The results indicate that GRQA is effective in general semantic segmentation as well and functions as a plug-and-play module.
> * **Suggestion for clarification regarding the Citation of Prototypes and Writing Errors:** In response to the reviewer’s suggestion, we add the recommended citation to related work and correct the writing errors in the manuscript.
>
> 4. We sincerely thank *Reviewer `dtcA`* for constructive feedback, insightful questions, and valuable suggestions. We have addressed and resolved the reviewers’ concerns as follows.
> * **Concern about Applicability to High-Quality, Non-Real-Time Models:** Following the reviewer’s suggestion, we apply GRQA to state-of-the-art DGSS methods, REIN and SoMA (both high-quality, non-real-time models), achieving improvements of 1.2 and 0.6, respectively. The results demonstrate that GRQA has broad applicability and functions as a plug-and-play module.
> * **Request for More Real-Time Semantic Segmentation (RT-SS) Methods baseline:** Following the reviewer’s request, we compare our method against additional real-time segmentation models such as SeaFormer, RTFormer, and Next-ViT. The results show that our method outperforms these models in domain generalization while achieving a balanced trade-off between speed and performance.
> * **Request for Evaluation under Corruptions Setting:** In response to the reviewer’s request, We evaluate our method on Cityscapes-C and achieve 69.8 mIoU and 54 FPS, outperforming other models in both speed and accuracy. The results demonstrate that our method maintains strong generalization under corruption settings, while also achieving real-time inference speed.
> * **Suggestion for Justifying Efficiency Benefits (Parameter Count and Inference Speed):** Following the reviewer’s suggestion, we analyze our method’s parameters, speed, and domain generalization, showing a 5× speed improvement and a 10M reduction in parameters compared to Mask2Former. These results highlight our method's advantages in parameter efficiency, inference speed, and domain generalization.

---

### Meta-Review · Area_Chair_cmtF · 2026-01-06

**Summary:**

In summary, the reviewers agreed that the problem of Real-Time Domain-Generalized Semantic Segmentation (RT-DGSS) is novel, practical, and well-motivated. While there were some concerns, I believe the authors provided a comprehensive rebuttal that addressed these issues effectively. Also, I believe the GRQA is an interesting component and can inspire future works in the field of semantic segmentation with real-time reasoning. Therefore, I recommend accepting the paper. More specifically, below are some main concerns.

**Baselines and Comparisons**:
Reviewer dtcA (who gave the lower score 4) and Reviewer Zb67 requested more rigorous comparisons. I think the authors added all the requested comparisons, especially directly addressed the lower-scoring reviewer's main comment.

**Technical Justification for Efficiency**:
A primary question from the reviewer 9SAK, where the authors performed a thorough ablation on Mask2Former, which proved the decoder is the bottleneck of latency.

**Generalizability**:
Reviewers dtcA and fRjv asked if the GRQA module acts as a "plug-and-play" solution, and the authors showed that GRQA improves performane when added to other models like REIN and SoMA (even though the improvements are relatively small).

**Reviewer Concerns:**

I think all reviewers' concerns have been adequately addressed by the rebuttal.

**Reviewer Scores:**

**Zb67**: The reviewer will probably keep the positive rating 6 or increase it to 8

**9SAk**: The reviewer will probably keep the positive rating 6 or increase it to 8

**fRjv**: The reviewer will probably keep the positive rating 6 or increase it to 8

**dtcA**: The reviewer will probably increase the rating to 8

---

### Decision · Program_Chairs · 2026-01-26

Accept (Poster)